# *Naa12* compensates for *Naa10* in mice in the amino-terminal acetylation pathway

Hyae Yon Kweon[1†], Mi-Ni Lee[1,2†‡]*, Max Dorfel[3], Seungwoon Seo[1],
Leah Gottlieb[4,5], Thomas PaPazyan[3], Nina McTiernan[6], Rasmus Ree[6],
David Bolton[7], Andrew Garcia[8], Michael Flory[9], Jonathan Crain[3], Alison Sebold[3],
Scott Lyons[3], Ahmed Ismail[3], Elaine Marchi[8], Seong-keun Sonn[1], Se-Jin Jeong[10],
Sejin Jeon[1], Shinyeong Ju[11], Simon J Conway[12], Taesoo Kim[1], Hyun-Seok Kim[1],
Cheolju Lee[11,13], Tae-Young Roh[14], Thomas Arnesen[6,15,16],
Ronen Marmorstein[4,5,17], Goo Taeg Oh[1]*, Gholson J Lyon[3,8,18,19]*

[1]Department of Life Science and College of Natural Sciences, Ewha Womans University, Seoul, Republic of Korea; [2]Laboratory Animal Resource Center Korea ResearchInstitute of Bioscience and Biotechnology, Chungbuk, Republic of Korea; [3]Stanley Institute for Cognitive Genomics, Cold Spring Harbor Laboratory, Woodbury, United States; [4]Department of Chemistry, University of Pennsylvania, Philadelphia, United States; [5]Abramson Family Cancer Research Institute, Perelman School of Medicine, University of Pennsylvania, Philadelphia, United States; [6]Department of Biomedicine, University of Bergen, Bergen, Norway; [7]Department of Molecular Biology, New York State Institute for Basic Research in Developmental Disabilities, Staten Island, United States; [8]Department of Human Genetics, New York State Institute for Basic Research in Developmental Disabilities, Staten Island, United States; [9]Research Design and Analysis Service, New York State Institute for Basic Research in Developmental Disabilities, Staten Island, United States; [10]Center for Cardiovascular Research, Washington University School of Medicine, Saint Louis, United States; [11]Center for Theragnosis, Korea Institute of Science and Technology, Seoul, Republic of Korea; [12]Herman B. Wells Center for Pediatric Research, Indiana University School of Medicine, Indianapolis, United States; [13]Department of Converging Science and Technology, KHU-KIST, Kyung Hee University, Seoul, Republic of Korea; [14]Department of Life Sciences, Pohang University of Science and Technology, Pohang, Republic of Korea; [15]Department of Biological Sciences, University of Bergen, Bergen, Norway; [16]Department of Surgery, Haukeland University Hospital, Bergen, Norway; [17]Department of Biochemistry and Biophysics, Perelman School of Medicine, University of Pennsylvania, Philadelphia, United States; [18]Biology PhD Program, The Graduate Center, The City University of New York, New York, United States; [19]George A. Jervis Clinic, New York State Institute for Basic Research in Developmental Disabilities, Staten Island, United States

*For correspondence:
minilee@kribb.re.kr (M-NL);
gootaeg@ewha.ac.kr (GTO);
gholsonjlyon@gmail.com (GJL)

†These authors contributed equally to this work

Present address: ‡Laboratory Animal Resource Center, Korea Research Institute of Bioscience and Biotechnology, Chungbuk, Republic of Korea

Competing interests: The authors declare that no competing interests exist.

**Abstract** Amino-terminal acetylation is catalyzed by a set of N-terminal acetyltransferases (NATs). The NatA complex (including X-linked Naa10 and Naa15) is the major acetyltransferase, with 40–50% of all mammalian proteins being potential substrates. However, the overall role of amino-terminal acetylation on a whole-organism level is poorly understood, particularly in mammals. Male mice lacking *Naa10* show no globally apparent in vivo amino-terminal acetylation impairment and do not exhibit complete embryonic lethality. Rather *Naa10* nulls display increased neonatal lethality, and the majority of surviving undersized mutants exhibit a combination of

hydrocephaly, cardiac defects, homeotic anterior transformation, piebaldism, and urogenital anomalies. *Naa12* is a previously unannotated *Naa10*-like paralog with NAT activity that genetically compensates for *Naa10*. Mice deficient for *Naa12* have no apparent phenotype, whereas mice deficient for *Naa10* and *Naa12* display embryonic lethality. The discovery of *Naa12* adds to the currently known machinery involved in amino-terminal acetylation in mice.

## Introduction

Amino-terminal acetylation is one of the most common protein modifications, occurring co- and post-translationally. Approximately 80% of cytosolic proteins are amino-terminally acetylated in humans and ~50% in yeast (*Arnesen et al., 2009*), while amino-terminal acetylation is less common in prokaryotes and archaea (*Dörfel and Lyon, 2015*). Amino-terminal acetylation is catalyzed by a set of enzymes, the N-terminal acetyltransferases (NATs), which transfer an acetyl group from acetyl-coenzyme A (Ac-CoA) to the free α-amino group of a protein's N-terminus. To date, eight distinct NATs (NatA–NatH) have been identified in eukaryotes that are classified based on different subunit compositions and substrate specificities (*Polevoda et al., 2009*; *Aksnes et al., 2019*; *Starheim et al., 2012*). Amino-terminal acetylation has been implicated in steering protein folding, stability or degradation, subcellular targeting, and complex formation (*Ree et al., 2018*; *Shemorry et al., 2013*; *Dikiy and Eliezer, 2014*; *Holmes et al., 2014*; *Scott et al., 2011*). The vital role of NATs and amino-terminal acetylation in development has also emerged (*Lee et al., 2018*).

NatA, the major NAT complex, targets ~40% of the human proteome, acetylating Ser-, Ala-, Gly-, Thr-, Val-, and Cys N-termini after removal of the initiator methionine (*Arnesen et al., 2009*; *Starheim et al., 2012*). Human NatA consists of two main subunits, the catalytic subunit N-α-acetyl-transferase 10 (NAA10) (Ard1) and the auxiliary subunit NAA15 (Nat1), and a regulatory subunit HYPK (*Arnesen et al., 2005*; *Arnesen et al., 2010*; *Gottlieb and Marmorstein, 2018*). NAA15 function has been linked to cell survival, tumor progression, and retinal development (*Arnesen et al., 2006a*; *Gendron et al., 2010*). In addition, Naa10-catalyzed N-terminal acetylation has been reported to be essential for development in many species (*Lee et al., 2018*; *Wang et al., 2010*; *Chen et al., 2014*; *Linster et al., 2015*; *Ree et al., 2015*; *Feng et al., 2016*; *Chen et al., 2018*), and although NatA is not essential in *Saccharomyces cerevisiae*, depletion of *Naa10* or *Naa15* has strong effects, including slow growth and decreased survival when exposed to various stresses (*Mullen et al., 1989*; *Polevoda and Sherman, 2003*).

*NAA10* mutations were found to be associated with several human diseases characterized by severe phenotypes, including global developmental defects (*Lee et al., 2018*). Among these, the X-linked Ogden syndrome (OS) (*Myklebust et al., 2015*; *Rope et al., 2011*) shows the most severe pathological features such as infant lethality and has reduced NatA catalytic activity. In a *S. cerevisiae* model for the Naa10 Ser37Pro mutant, the mutation impairs NatA complex formation and leads to a reduction in NatA catalytic activity and functionality (*Van Damme et al., 2014*; *Dörfel et al., 2017*). Further, OS patient-derived cells have impaired amino-terminal acetylation in vivo of some NatA substrates (*Myklebust et al., 2015*). Over the years, many additional pathogenic *NAA10* variants have been identified in *NAA10* or NAA15 (*Esmailpour et al., 2014*; *Popp et al., 2015*; *Casey et al., 2015*; *McTiernan et al., 2018*; *Ree et al., 2019*; *Støve et al., 2018*; *Cheng et al., 2019*; *Cheng et al., 2018*; *Johnston et al., 2019*) and the collection of presenting symptoms for families with *NAA10* mutations is currently referred to as Ogden syndrome or *NAA10*-related syndrome (*Wu and Lyon, 2018*).

The autosomal *NAA10* homolog, *NAA11* (ARD2), has been reported to be present in mice and humans, and is co-expressed with *NAA10* in human cell lines (*Arnesen et al., 2006b*). Therefore, *NAA11* could conceivably compensate when *NAA10* is reduced or lacking (*Lee et al., 2018*). However, *NAA11* was only found in testis and placenta in human and gonadal tissues in mouse, while *NAA10* showed widespread expression in various tissues in embryos and adults (*Pang et al., 2011*). Thus, any functional redundancy or compensation might be limited to certain tissues only.

To elucidate the functional role of *Naa10* during development in mice, we used two different *Naa10*-deficient mouse lines: one, referred to as *Naa10* knockout (KO), which was previously reported specifically related to bone density in postnatal day 3 (P3) mice (*Yoon et al., 2014*), and another denoted as *Naa10*$^{tm1a(EUCOMM)Hmgu}$ (*Naa10*$^{tm1a}$), generated in this study. These *Naa10*-

deficient mice exhibit pleiotropic developmental abnormalities at a range of different ages, overlapping with some of the phenotypes seen in human disease involving NAA10 impairment. Because we did not discover major changes in the overall Nt-acetylome in *Naa10* KO mice, we hypothesized that there might be a compensating gene in mice, which led us to the identification of a new paralog of *Naa10,* which we name *Naa12. Naa12* is expressed in several organs (liver, kidney, heart, and testis) and, like Naa10, binds to Naa15 to mediate NatA activity. Furthermore, lethality was observed in *Naa10 Naa12* double-KO mice, which supports the compensatory role of *Naa12* in vivo. Thus, we demonstrate that *Naa10* is essential for proper development and *Naa12*, a newly identified paralog of *Naa10*, can play a compensatory role in mice.

## Results

### *Naa10* KO mice can be born, but display pleiotropic developmental defects

To explore the role of *Naa10* in development, most analyses were carried out using our *Naa10* KO model mice that had been generated previously (*Yoon et al., 2014*) using a targeting vector deleting Exon1, including the start codon, and Exon2 to Exon4 containing the GNAT domain including the acetyl-CoA binding motif, which is crucial for *Naa10* function. We also generated another *Naa10*-deficient mouse, which we called *Naa10^tm1a^*, expressing β-galactosidase rather than the *Naa10* gene (*Figure 1—figure supplement 1A*). *Naa10* expression was deficient in *Naa10^tm1a^* mice (*Figure 1—figure supplement 1B, C*). Especially strong β-gal staining was observed during embryonic stages in the brain, heart, and spinal cord (*Figure 1—figure supplement 1D*). Male *Naa10* KO (*Naa10^-/Y^*) embryos displayed mild to severe developmental defects compared to wildtype (WT) (*Naa10^+/Y^*) embryos. Some *Naa10^-/Y^* mice had lower levels of somites and developmental delay. Additionally, some *Naa10^-/Y^* embryos had a normal number of somites but were retarded in growth (*Figure 1A*). Some of the embryos underwent lysis or remained arrested at an earlier stage than embryonic day 10.5 (E10.5), with no turning, an abnormal trunk, and underdeveloped facial features. These phenotypes also reproduced in *Naa10^tm1a/Y^* embryos. Next, we assessed whether *Naa10* is essential for viability and counted the Mendelian ratios. Both *Naa10^-/Y^* and *Naa10^tm1a/Y^* mice were under-represented after birth, while there was no significant reduction in the embryonic stage in both mouse lines (*Supplementary file 1a, b*). We monitored the pups daily at postnatal day 0 (P0) to postnatal day 3 (P3) and beyond, and the survival rate of *Naa10^-/Y^* mice dramatically decreased relative to either WT (*Naa10^+/Y^* and *Naa10^+/+^*) or heterozygous female (*Naa10^+/-^*) mice after the first few days of life (*Figure 1B*), and a few *Naa10^-/Y^* mice with postnatal lethality exhibited severe developmental defects such as craniofacial anomaly, an undeveloped lower body, whole-body edema, and ocular malformations (*Figure 1C*).

Congenital heart defects are one of the main causes of infant lethality, and cardiac diseases are a common developmental anomaly in OS patients (*Casey et al., 2015*), with some OS males dying in infancy with cardiac arrhythmias (*Rope et al., 2011*). Therefore, we investigated whether *Naa10* KO affects cardiac development. Development of a four-chambered septated heart is normally complete at E14.5; therefore, we examined the cardiovascular system at E14.5. We identified ventricular septal defects (VSDs) in several *Naa10^-/Y^* embryos, as well as concomitant double outlet right ventricle (DORV) at E14.5 (*Figure 1D*, upper). VSDs and atrial septal defects (ASDs) were also observed at E18.5 (*Figure 1d*, bottom), and persistent truncus arteriosus (PTA) or DORV, along with concomitant membranous and muscular VSDs, were found in several of the mice that died in the first day of life (n = 6/28 examined). Given the presence of outflow tract defects and VSDs, we examined whether the ductus arteriosus had closed appropriately or not at birth. Significantly, both *Naa10^-/Y^* and *Naa10^-/-^* females (n = 3/28 examined) exhibited a patent ductus arteriosus, meaning that there is a failure of the mutant in utero cardiovascular system to adapt to adult life (birth) and close the interatrial and aorta-pulmonary trunk shunts that are required for normal fetal life (*Conway et al., 2003*). As murine outflow tract and VSDs are not compatible with postnatal survival (*Conway et al., 2003*), these data suggest that congenital heart defects in *Naa10^-/Y^* mice may explain some of their neonatal lethality (*Figure 1—figure supplement 2*). We also examined surviving adult mice for any possible situs inversus, but we did not observe this in any adult (>4 weeks) *Naa10^-/Y^* mice examined (n = 19). Combined, these data suggest that *Naa10* mutant CHDs are mainly confined to aberrant

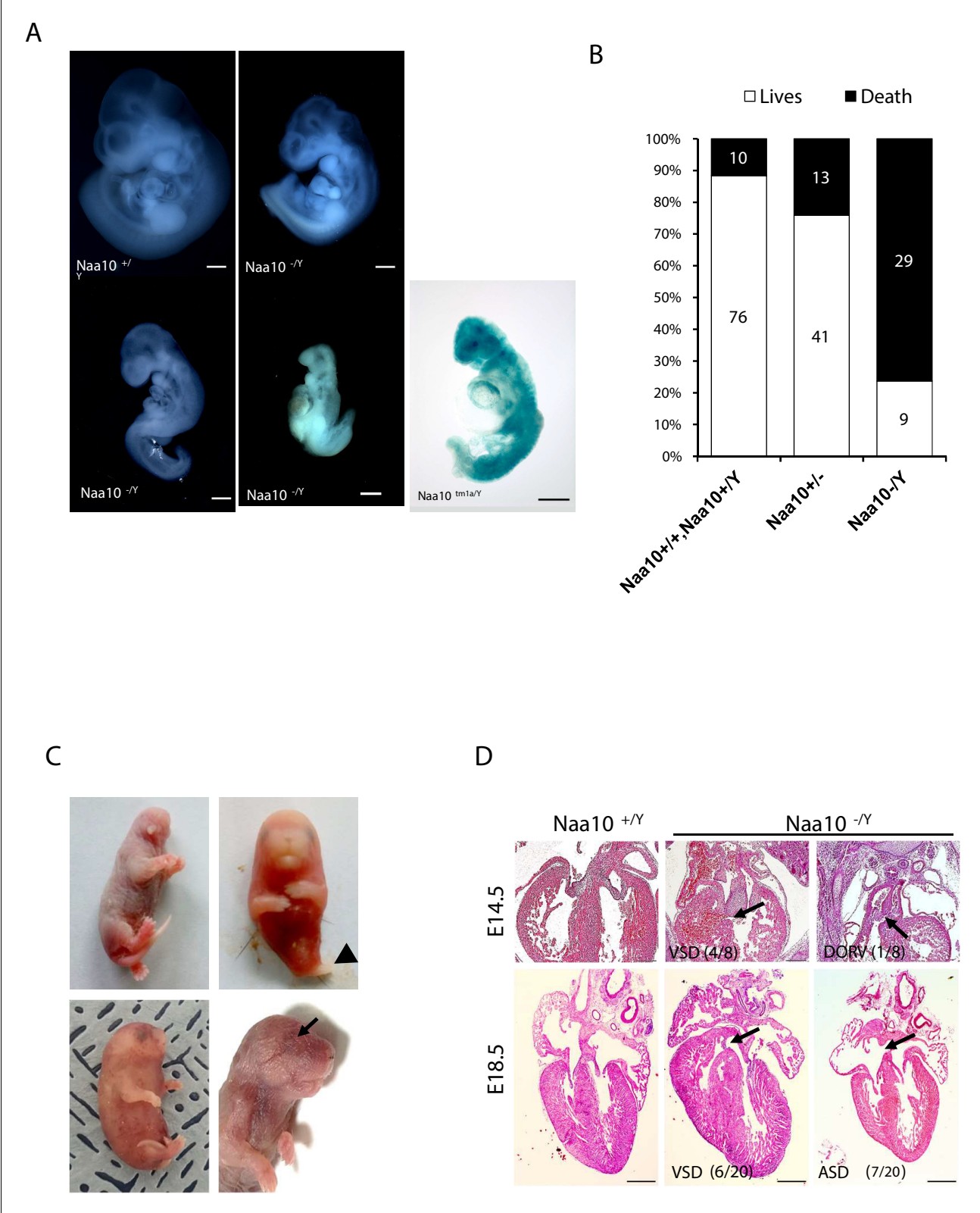

**Figure 1.** Deficiency of *Naa10* leads to abnormal development and postnatal lethality. (**A**) *Naa10⁺/Y*, *Naa10⁻/Y*, and *Naa10^{tm1a/Y}* embryos at E10.5. Growth retardation (5/33, more than five somites lower or undersized compared to littermate controls), kinky trunk, and developmental arrest are shown in *Naa10⁻/Y* (4/33) and *Naa10^{tm1a/Y}* (1/5). Scale bars: 500 µm. (**B**) The percentage lethality in newborns, comparing *Naa10* wildtype (WT) (*Naa10⁺/Y* and *Naa10⁺/⁺*), *Naa10⁻/⁺* and *Naa10⁻/Y* pups until P3, derived from matings between heterozygous females and WT males. Approximately 11.6% (10/86) of
*Figure 1 continued on next page*

*Figure 1 continued*

WT, 24% (13/54) of *Naa10+/-*, and 76.3% (29/38) of *Naa10-/Y* mice were found dead before P3. (**C**) Representative images of *Naa10-/Y* pups during early postnatal days compared with *Naa10+/Y*. Severe developmental defects such as malformations of head and lower body (one leg; black arrowheads), whole-body edema, and anophthalmia (black arrows) are shown (N = 1 each). (**D**) Hematoxylin and eosin (H&E)-stained heart transverse section at E14.5 and vertical section at E18.5, comparing *Naa10+/Y* and *Naa10-/Y* embryos. *Naa10-/Y* embryo shows a ventricular septal defect (VSD) at E14.5 and E18.5. Also, at E18.5, *Naa10-/Y* embryo shows atrial septal defect (ASD). Arrow indicates VSD, ASD, and double outlet right ventricle (DORV). Scale bars: 20 μm.

The online version of this article includes the following source data and figure supplement(s) for figure 1:

**Figure supplement 1.** Generation and confirmation of Naa10tm1a mice.

**Figure supplement 1—source data 1.** Generation and confirmation of Naa10tm1a mice.

**Figure supplement 1—source data 2.** Generation and confirmation of Naa10tm1a mice.

**Figure supplement 1—source data 3.** Generation and confirmation of Naa10tm1a mice.

**Figure supplement 2.** Gross anatomy and histology of neonatal mouse hearts.

remodeling of the great vessels of the heart, leading to pulmonary overload at birth resulting in lethality.

Some of the surviving homozygous mice (*Naa10-/Y* and *Naa10-/-*) had reduced body weight (*Figure 2A*). This reduced body weight continued through weaning, and some mice lost more weight as they developed progressive hydrocephaly. We observed that the smallest weight animal between the *Naa10+/Y* and one *Naa10-/Y* genotypes was almost always the *Naa10-/Y* genotype when the analysis was restricted to only include litters in which there was at least one of each of those genotypes living beyond 4 days of life. For example, 13 litters met this criteria from the mating (*Naa10+/- × Naa10+/Y*), and 12/13 of the litters had the *Naa10-/Y* as the lower weight (Fisher's exact test, two-tailed, p-value<0.0001). Five litters met this criteria from the mating (*Naa10+/- × Naa10-/Y*), and of these, all of them had the *Naa10-/Y* as the lower weight (Fisher's exact test, two-tailed, p-value=0.0079). Therefore, despite the known variability in weight data as a function of genetic background, environment, and stochastic variation (*Pun et al., 2013*), it does appear at least for 'within-litter' analysis that *Naa10-/Y* males are born at a smaller weight than *Naa10+/Y* males and on average remained the smallest male in the litter throughout their life.

Although piebaldism has never been reported in humans with OS, all (100%) of the *Naa10-/Y* and *Naa10tm1a/Y* mice exhibited hypopigmentation on their belly (*Figure 2B*, upper), with this piebaldism quite varied in its extent but not appearing to correlate in any way with other phenotypes, such as hydrocephaly. Another phenotype with complete penetrance was bilateral supernumerary ribs (14 pairs of ribs instead of 13) in all *Naa10-/Y* and *Naa10tm1a/Y* mice (*Figure 2B*, middle and bottom, *Table 1*). This extra pair of ribs linking to the sternum transforms the T8 vertebrae into an anterior T7-like phenotype (*Figure 2—figure supplement 1*, *Table 1*).

A majority of the *Naa10-/Y* and *Naa10-/-* mice also had four instead of the usual three sternebrae, which were sometimes fused (*Table 1*). Cervical vertebrae fusion was also demonstrated in *Naa10-/Y* mice, particularly involving C1 and C2, suggesting possible anteriorization of C2 into a C1-like phenotype (*Figure 2—figure supplement 1E, F*, *Supplementary file 1c*). The number of lumbar vertebrae remained the same, thus suggesting an anterior transformation of the first sacral vertebra to a lumbar-like phenotype. These combined observations suggest possible anterior transformations in the *Naa10* mutant skeletal phenotype, with an anteriorization of C2, a T8 transformation to a T7-like phenotype with ribs connecting to the sternum, an extra pair of ribs on L1 likely due to an L1 transformation to a T13-like phenotype, and an anterior transformation of the first sacral vertebra to a lumbar-like phenotype with loss of fusion to the sacral wings.

Out of 32 *Naa10-/Y* that survived past the third day of life and which were then examined longitudinally, about 60% survived past 200 days of life (~7 months) (*Figure 2—figure supplement 2*), with some of these then developing hydronephrosis (*Figure 2C*, middle). They had some hollowed space in the kidney, which had been filled with fluid and their ureter was thickened already at P3 stage of prenatal development in some *Naa10-/Y* mice (*Figure 2—figure supplement 2A*). Commonly, hydronephrosis is caused by a blockage or obstruction in the urinary tract. We speculate that this swelling in *Naa10* KO (*Naa10-/Y* and *Naa10-/-*) mice is likely caused by ureteral defects rather than the kidney itself. Moreover, some *Naa10* KO mice displayed genital defects, such as seminal vesicle malformation and hydrometrocolpos, respectively (*Figure 2C*, bottom). Many *Naa10-/-* female mice appeared

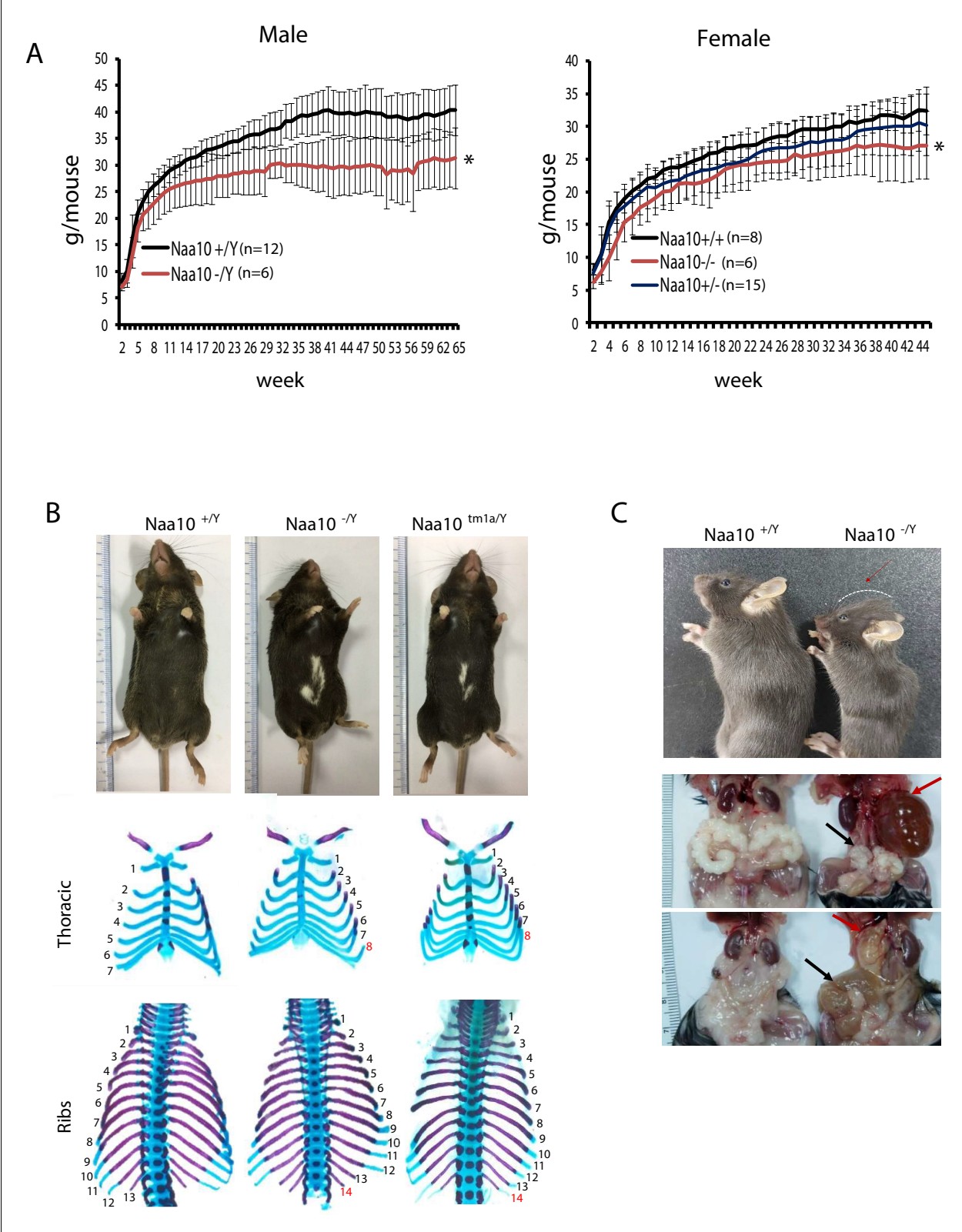

**Figure 2.** Pleiotropic phenotypes of *Naa10* knockout (KO) mice. (**A–C**) Representative images of abnormalities in *Naa10⁻/Y* compared with *Naa10⁺/Y*. (**A**) Body weight of male (left) and female (right) versus ages was monitored from 2 weeks. The weight of *Naa10⁻/Y* and *Naa10⁻/⁻* mice is markedly reduced compared with that of the wildtype (WT) mice. Asterisks indicate a statistical difference calculated by Student's *t*-test: *p<0.05. (**B**) Representative images of completely penetrant phenotypes. Hypopigmentation (*Naa10⁺/Y*, n = 243; *Naa10⁻/Y*, n = 121; *Naa10^{tm1a/Y}*, n = 17) and supernumerary ribs
*Figure 2 continued on next page*

*Figure 2 continued*

($Naa10^{+/Y}$, n = 3; $Naa10^{-/Y}$, n = 6; $Naa10^{tm1a/Y}$, n = 2; E18.5) were found 100% in $Naa10$-deficient mice. (C) $Naa10^{-/Y}$ is smaller in size and has round-shaped head ($Naa10^{+/Y}$ 0/59, $Naa10^{-/Y}$ 7/33). Over time, hydrocephaly became apparent (N = 14/29 [~48%] for >P7 male $Naa10^{-Y}$; N = 7/19 [~36%] for >P7 female $Naa10^{-/}$). Hydronephrosis (red arrow, $Naa10^{+/Y}$ 0/23, $Naa10^{-/Y}$ 14/29, $Naa10^{+/+}$ 0/5, $Naa10^{-/-}$ 7/19) and abnormal genitalia (black arrow) of male (middle, $Naa10^{+/Y}$ 0/23, $Naa10^{-/Y}$ 16/29) and female (bottom, hydrometrocolpos, $Naa10^{+/+}$ 0/5, $Naa10^{-/-}$ 7/19) are shown.

The online version of this article includes the following figure supplement(s) for figure 2:

**Figure supplement 1.** Skeletal phenotype by CT scanning.

**Figure supplement 2.** Hydronephrosis and hydrocephaly in Naa10 KO mice.

to have decreased fecundity, although they were fertile upon the first mating, and this decrease in fecundity is possibly due to the development of hydrometrocolpos (*Figure 2C*, bottom), which might result from structural issues, like vaginal atresia or a retained vaginal septum, although this requires further investigation. Additionally, hydrocephaly became clinically apparent with a round-shaped head (*Figure 2C*, upper) in ~40% of the $Naa10^{-/Y}$ mice that had survived past 3 days of life (*Figure 2—figure supplement 2C*). CT scanning of some of these mice confirmed hydrocephaly as the primary cause of their rapid deteriorating condition, usually within the first three months of life (*Figure 2—figure supplement 2B, C*). CT scanning did not reveal any obstructive lesions (such as a tumor) in any of the ventricles that could account for the hydrocephaly. Taken together, these results indicate that $Naa10$ contributes to overall development and is particularly important for viability.

Litter sizes and offspring from other matings were also investigated, as shown in *Supplementary file 1d*. Matings were setup between $Naa10^{-/-}$ females and C57bl6J WT ($Naa10^{+/Y}$) males, involving 11 mating pairs with seven unique females and seven unique males. Of a total of 127 pups that were born, 37 died in the first day of life and were degraded and/or cannibalized prior to any tail sample being retrieved, thus not being genotyped. This was a relatively high death rate in the first 24 hr of life (29%), more so than with the other matings, except for the one between $Naa10^{-/-}$ females and $Naa10^{-/Y}$ males (*Supplementary file 1d*). However, this is substantially less than the death rate of 90% (46/51) reported for the same mating in the *Lee et al., 2017* paper, and we currently do not have an explanation for this discrepancy. Of the remaining 90 pups that could be genotyped, 59 of these were $Naa10^{+/-}$ females and 31 were $Naa10^{-/Y}$ males. 7 of the 59 $Naa10^{+/-}$ females and 2 of the 31 $Naa10^{-/Y}$ males died in the first three days of life (for a total death rate in the first three days for all born pups of 46/127, or 36%), and after this time, none of the remaining $Naa10^{+/-}$ females died in the first 10 weeks of life (52/59, or 88% overall survival), whereas 10 of the remaining 29 $Naa10^{-/Y}$ males developed hydrocephaly and died in the first 10 weeks of life, for an overall survival of (19/31, or 61%). The death rate for all pups of 36% in the first three days of life is similar to the rate of 42.4% seen with the mating of $Naa10^{-/-}$ females with $Naa10^{-/Y}$ males

**Table 1.** Skeletal analyses for ribs, sternebrae, and vertebrae.

| | $Naa10^{+/Y}$ (n = 50) | $Naa10^{+/+}$ (n = 10) | $Naa10^{+/-}$ (n = 17) | $Naa10^{-/Y}$ (n = 17) | $Naa10^{-/-}$ (n = 1) |
|---|---|---|---|---|---|
| 4 sternebrae | 7 (14.0%) | 1 (10%) | 3 (17.6%) | 9 (52.9%) | 1 (100%) |
| 3 sternebrae | 27 (54.0%) | 8 (80%) | 11 (64.7%) | 5 (29.4%) | 0 (0%) |
| 4 sternebrae but with 3/4 fusion | 16 (32%) | 1 (10%) | 3 (17.6%) | 3 (17.6%) | 0 (0%) |
| 14 ribs total bilaterally | 0 (0%) | 0 (0%) | 0 (0%) | 17 (100%) | 1 (100%) |
| 13 ribs total bilaterally | 50 (100%) | 10 (100%) | 17 (100%) | 0 (0%) | 0 (0%) |
| 8 ribs attached to sternum bilaterally | 0 (0%) | 0 (0%) | 0 (0%) | 17 (100%) | 1 (100%) |
| 7 ribs attached to sternum bilaterally | 50 (100%) | 10 (100%) | 17 (100%) | 0 (0%) | 0 (0%) |
| 14 thoracic vertebrae | 0 (0%) | 0 (0%) | 0 (0%) | 17 (100%) | 1 (100%) |
| 13 thoracic vertebrae | 50 (100%) | 10 (100%) | 17 (100%) | 0 (0%) | 0 (0%) |

Tabulation regarding the number of sternebrae found in skeletons, including ones in which there was partial fusions between the third and fourth sternebrae.

(*Supplementary file 1d*), whereas this rate is higher than that seen for *Naa10⁺/⁻* females mated with *Naa10⁺/Y* males (15.8%) or with *Naa10⁻/Y* males (13.6%).

### *Naa10*-deficient mice have a functionally active NatA complex

Prior experiments showed reduced in vivo protein amino-terminal acetylation of a few putative targets in patient cells (*Myklebust et al., 2015*). Reduced Nt-acetylomes were also observed in the *Naa10* mutant yeast models (*Van Damme et al., 2014*). Given these prior reports, we hypothesized that pleiotropic phenotypes in *Naa10*-deficient mice are due to a decrease in global N-terminal acetylation. To test our hypothesis, integrated N-terminal peptide enrichment method (iNrich) (*Ju et al., 2020*) was used to analyze the level of protein amino-terminal acetylation in mouse embryonic fibroblast (MEF) lysates of *Naa10⁺/Y* and *Naa10⁻/Y*. Since the samples are treated with deuterated acetic anhydride prior to MS, unacetylated N-terminal site appears with +3 Da mass shift in the MS spectrum of the corresponding acetylated N-terminal site (*Van Damme et al., 2011a*). The peak intensity ratios of acetyl/heavy acetyl pairs represent the degree of acetylation of the N-terminal site. We found 765 acetyl/heavy acetyl pairs of N-termini throughout five replicates of *Naa10⁺/Y* and five replicates of *Naa10⁻/Y* MEFs. Except for the sites detected only in either WT or mutant, 533 N-terminal sites could be compared (see tabs called 'N-term' and 'Header Key' in *Supplementary file 2a, c*). Approximately 98% (n = 522) of N-termini sites showed less than 10% variation in the degree of terminal acetylation, indicating that there is no major difference in amino-terminal acetylation between *Naa10⁻/Y* and *Naa10⁺/Y* MEFs (*Figure 3A*). A more stringent analysis was also conducted in which peptides had to be detected in all 10 samples (i.e., tabs marked 'N-term detected in all samples'

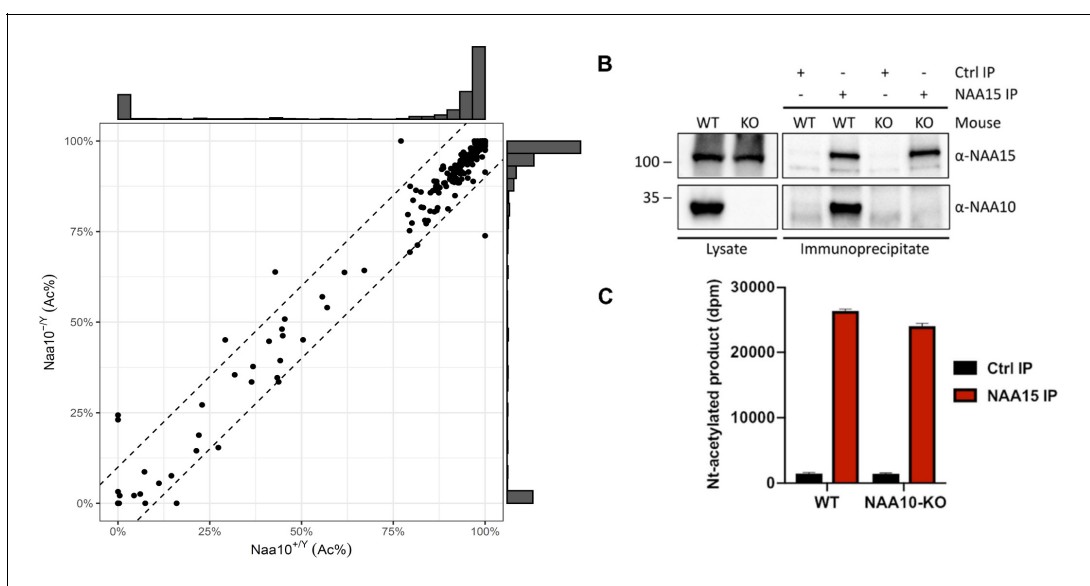

**Figure 3.** Activity measurement of NatA from wildtype (WT) and *Naa10* knockout (KO) mice. (**A**) Correlation of Naa10 alteration state on amino-terminal acetylation in mouse embryonic fibroblasts (MEFs). Each dot (n = 533) represents the average amino-terminal acetylation percentage of five replicates of *Naa10⁺/Y* and *Naa10⁻/Y*, respectively. Dashed lines are the borders of ±10% difference. Except for the 10 dots, 522 of the 533 dots are within the borders. The marginal histograms show the distribution of amino-terminal acetylation data points. (**B**) Immunoprecipitation of Naa15. Liver tissue from WT and *Naa10* KO mouse was lysed and incubated with anti-Naa15 antibody to retrieve NatA complexes. Proteins were separated by SDS-PAGE and immunoblots probed with anti-Naa15 antibody and anti-NAA10 antibody. (**C**) Catalytic activity of immunoprecipitated NatA. The catalytic activity of NatA precipitated from WT and *Naa10* KO mouse liver tissue by anti-Naa15 was measured towards the NatA substrate peptide SESS$_{24}$ in an in vitro [$^{14}$C]-Ac-CoA–based acetylation assay. Control reactions were performed with no enzyme or no peptide to account for background signal. The immunoprecipitation (IP) and activity measurements were performed in three independent setups, each with three technical replicates per assay. One representative setup is shown.

The online version of this article includes the following source data and figure supplement(s) for figure 3:

**Source data 1.** Identification of a potential Naa10 homolog.

**Figure supplement 1.** Identification of a potential Naa10 homolog.

**Figure supplement 1—source data 1.** Identification of a potential Naa10 homolog.

and 'Header Key' in *Supplementary file 2b, c*), and this resulted in 152 N-termini sites, of which only 3 (Rpl27, PPia, and Histone H1.0) had a slightly greater than 10% difference in the degree of acetylation between $Naa10^{+/Y}$ and $Naa10^{-/Y}$. Although this was not a significant result statistically (p=0.09), it is worth noting that peptidyl-prolyl cis–trans isomerase A (PPIA), having a 10.3% decrease in amino-terminal acetylation, was previously identified with decreased amino-terminal acetylation in patient-derived B cells and fibroblasts in boys with the S37P mutation in NAA10 (*Myklebust et al., 2015*), along with being decreased in siNatA knockdown HeLa cells (*Arnesen et al., 2009*). PPIA also had decreased amino-terminal acetylation in one sample from homozygous null $NAA15^{L314*/L314*}$-induced pluripotent stem cells (*Ward et al., 2021*).

Overall, given the very minor differences with amino-terminal acetylation, we measured the in vitro amino-terminal acetylation activity of NatA via immunoprecipitation of the large auxiliary sub-unit Naa15 from mouse tissues. This analysis showed normal expression of Naa15 in *Naa10* KO liver tissue as in WT tissues (*Figure 3B*), and we isolated a physical complex composed of Naa15 and undefined partners that retains NatA activity from *Naa10* KO tissues (*Figure 3C*). These data suggest that despite the loss of *Naa10* in mice the NatA complex remains active, thus explaining the lack of major differences with amino-terminal acetylation.

## A *Naa10* paralog exists in mice

*Naa10* disruption is lethal in a variety of organisms, including *Drosophila melanogaster* (*Wang et al., 2010*), *C. elegans* (*Chen et al., 2014*), and *Trypanosoma brucei* (*Ingram et al., 2000*). Given the relatively mild phenotype and no reduction of the Nt-acetylome in *Naa10* KO mice, we hypothesized that there might be a yet unidentified paralog of *Naa10,* which can compensate for loss of function in mice. A Blast search for genomic sequences with homology to *Naa10* exposed several *Naa10* pseudogenes on chromosomes 2, 3, 7, 12, 15, and 18. Additionally, Southern blot analysis from C57BL/6J DNA with *Naa10* cDNA probe detected bands of the expected sizes on the X chromosome (*Figure 3—figure supplement 1A, B*), while other bands of unexpected sizes appeared on other chromosomes 2, 5, 15, and 18. The previously identified *Naa10* paralog *Naa11* is located on chromosome 5; however, this paralog is only expressed in testes (*Pang et al., 2011*). We found a predicted gene (Gm16286, UniProt: Q9CQX6) on chromosome 18, with high similarity to *Naa10*, which we name *Naa12*, and RiboSeq and mRNA traces of this region suggest possible transcription and translation of this gene (*Figure 3—figure supplement 1C*). The protein sequence of *Naa12* is >80% identical to *Naa10* and almost 90% identical with *Naa11* (*Figure 4—figure supplement 1C*).

Quantitative PCR (q-PCR) analysis also confirmed the expression of this transcript in all tested tissues (*Figure 4—figure supplement 1A*), with the expression of Naa12 unchanged in the corresponding *Naa10* KO tissues. We attempted to test for Naa12 expression in mouse tissues by developing an antibody specific for Naa12 by performing a sequence alignment of the two known mNaa10 isoforms, mNaa11 and mNaa12, and selecting a unique Naa12 peptide for immunization and antibody generation (*Figure 4—figure supplement 1B*). After generation and affinity purification, we validated the specificity and sensitivity of this Naa12 antibody with recombinant proteins purified from bacterial hosts (*Figure 4—figure supplement 1C, D*). However, multiple attempts to use this antibody to detect Naa12 in mouse tissues met with conflicting results, so that we were unable to consistently detect Naa12 even in WT liver, kidney, or brain tissue lysates, which could be due to a poor antibody and/or very low expression or post-translational modification of Naa12 in these tissues, thus making it difficult to detect. Furthermore, given that this antibody was raised against a peptide at the C-terminus of Naa12, such data could not be used anyway to completely exclude the possibility of truncated non-functional mini-protein expression, although the lack of any signal with RT-PCR (*Figure 4—figure supplement 2*) likely means that nonsense-mediated decay occurred. Presently, the rabbit polyclonal antibody is no longer recognizing any consistent protein bands in western blotting, so we have abandoned any further attempts to use this antibody.

To test whether Naa12 has a similar enzymatic activity as Naa10, we performed a radioactive-based acetyltransferase assay using synthetic peptides (*Figure 4A*). Since monomeric Naa10 preferentially acetylates N-termini with acidic side chains (*Foyn et al., 2013*; *Van Damme et al., 2011b*; *Liszczak et al., 2013*), we used peptides representing the N-termini of γ-actin (starting DDDIA-) and γ-actin (starting EEEIA-), which are two known Naa10 in vitro substrates. Additionally, we used a peptide starting with SESSSKS-, representing an in vitro NatA complex substrate high-mobility

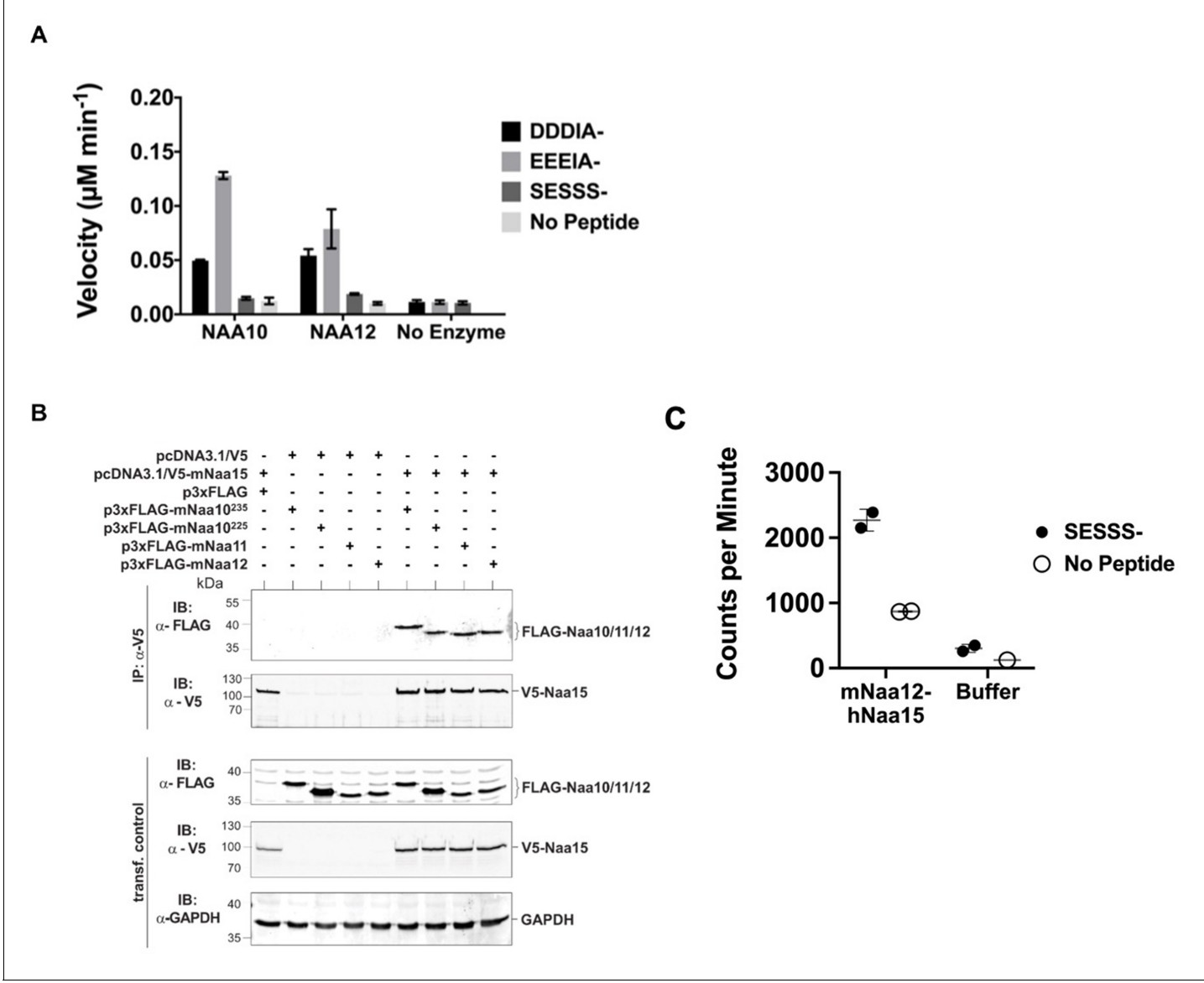

**Figure 4.** Characterization of *Naa12*. (**A**) In vitro N-terminal acetyltransferase radioactive-based assay. Comparison of mouse Naa10 and Naa12 towards Naa10 peptide substrates, beta-actin (DDDIA-) and gamma-actin (EEEIA-), and the optimal NatA complex peptide substrate, SESSS-. Background control reactions were performed in the absence of either peptide or enzyme. Assays were performed in triplicate; error bars represent SEM. (**B**) Co-immunoprecipitation assay. HEK293 cells were transfected as indicated and lysed after 48 hr. Cell lysates were incubated with 1 µg anti-V5 antibody to precipitate V5-tagged Naa15. The isolated complexes were separated on SDS-PAGE and probed with the indicated antibodies. (**C**) Recombinant mouse Naa12/human Naa15 chimera complex activity. Radioactive acetyltransferase activity assay evaluating the activity of mNaa12-hNaa15 towards peptide (closed circles, 'mNaa12-hNaa15') and peptide chemical acetylation in the absence of enzyme (closed circles, 'Buffer') as well as chemical acetylation of the enzyme in the absence of peptide (open circles) assay and background (open circles). Error bars represent SD of two technical replicates. These are the same results from fraction #14 (both SESSS- and No Peptide) and both Buffer and Background used to illustrate the size-exclusion-purified mNaa12-hNaa15 complex activity in *Figure 4—figure supplement 1F*.

The online version of this article includes the following source data, source code and figure supplement(s) for figure 4:

**Source data 1.** Characterization of a potential Naa10 homolog.
**Figure supplement 1.** Characterization of a potential Naa10 homolog.
**Figure supplement 1—source code 1.** Characterization of a potential Naa10 homolog.
**Figure supplement 1—source code 2.** Characterization of a potential Naa10 homolog.
**Figure supplement 1—source data 1.** Characterization of a potential Naa10 homolog.
**Figure supplement 1—source data 2.** Characterization of a potential Naa10 homolog.
**Figure supplement 2.** Confirmation and characterization of Naa12 knockout (KO) mice.

group protein A1. As expected for the monomeric proteins, we could not detect any activity towards the SESSSKS substrate. Importantly, both Naa10 and Naa12 significantly Nt-acetylated the acidic N-terminal peptides, demonstrating the intrinsic capacity of Naa12 to catalyze amino-terminal acetylation (*Figure 4A*).

Across species, Naa10 is bound to its auxiliary subunit, Naa15, which links the catalytic subunit to the ribosome to facilitate co-translational amino-terminal acetylation of proteins as they emerge from the exit tunnel (*Mullen et al., 1989*; *Sugiura et al., 2003*; *Park and Szostak, 1992*; *Gautschi et al., 2003*; *Magin et al., 2017*; *Varland and Arnesen, 2018*). Due to its high sequence similarity (*Figure 4—figure supplement 1B*), we suspected that Naa12 may also interact with Naa15. To test this hypothesis, we performed co-immunoprecipitation assays in HEK 293 cells. Apart from Naa10 (isoform 1, Naa10[235]) and Naa12, we also included the second isoform of mNaa10, mNaa10[225] that has been described earlier (*Arnesen et al., 2005*; *Park and Szostak, 1992*; *Kim et al., 2006*) as well as Naa11. Both Naa10 isoforms as well as Naa11 and Naa12 co-precipitated with V5-Naa15 but not V5 alone, suggesting that all tested proteins could form a stable complex with Naa15 in mouse (*Figure 4B*). As we have previously purified the human NatA complex composed of truncated human Naa10 (residues 1–160) and full-length human Naa15 complexes that had been expressed in insect cells (*Gottlieb and Marmorstein, 2018*), we attempted to co-express a chimeric truncated mouse Naa12 (residues 1–160) with full-length human Naa15 complex in insect cells (human and mouse Naa15 are highly conserved with a sequence conservation of 98.2%). The complex was purified by a combination of affinity, ion exchange, and size-exclusion chromatography, and size-exclusion fractions harboring a clearly detectable band of Naa15 and a lighter band for Naa12, as determined by silver staining, were analyzed for activity towards a SESSSKS- peptide (*Figure 4C*, *Figure 4—figure supplement 1E, F*). This analysis revealed that peak fractions containing the Naa12-Naa15 complex harbored detectable amino-terminal acetylation activity towards the SESSSKS- peptide (*Figure 4C*, *Figure 4—figure supplement 1F*), thus demonstrating catalytic activity of a NatA complex with mouse Naa12.

In a mass spectrometry analysis of a similar setup to that shown in *Figure 3B*, NAA15 immunoprecipitates from WT or *Naa10*-KO mouse livers were analyzed by mass spectrometry. We found five distinct peptides derived from Naa12 (*Table 2* and *Supplementary file 2d*). Three of these derive from the same part of the peptide sequence, RDLSQMADELRR, and all of these three peptides had one or two missed trypsin cleavages (DLSQMADELRR, RDLSQMADELR, and RDLSQMADELRR). The other two peptides, AMIENFSAK and ENQGSTLPGSEEASQQENLAGGDSGSDGK, are not the results of missed cleavages. None of these peptides are found in other sequences in the mouse genome and thus unambiguously identify Naa12 in our experiments. They have higher intensities in Naa15 IPs compared to Ctrl IPs, indicating that Naa12 is selectively enriched by Naa15 IP. Some peptides are additionally assigned to the Naa10/Naa11/Naa12 protein group as a large part of their sequences are identical. As expected, no unique Naa10 peptides are identified in the IPs from *Naa10*-KO mice. 12 peptides were ambiguously assigned to Naa12 or to major urinary proteins (Mup9, Mup8, Mup1, Mup17, Mup5, or Mup2), but these are as likely to be derived from Mups as from Naa12, as they have comparable intensities between Ctrl and Naa15 IPs.

### *Naa12* rescues loss of *Naa10* in mice

To investigate whether *Naa12* can rescue the loss of the function of Naa10 in vivo, *Naa12* KO mice were generated using CRISPR technology (*Singh et al., 2015*). One 95-base pair deletion Δ131–225 in *Naa12* was characterized in depth (*Figure 5A*). This mutation introduces a frameshift, leading to a termination codon at amino acid 67, which should either result in complete KO of the protein or, at best, the expression of a truncated mini-protein that would be far shorter than the usual 220 amino acid Naa12. We confirmed the deletion by PCR with genomic DNA (*Figure 5B*). QPCR further showed deletion of *Naa12* in the tested tissues of *Naa12* KO mice (*Figure 5C*); however, it seemed that *Naa12* might be slightly expressed in testis. Due to the high similarity between *Naa11* and *Naa12*, the expression shown in *Naa12* KO testis could actually be *Naa11* rather than *Naa12*, and this was confirmed by RT-PCR showing definite deletion (*Figure 4—figure supplement 2A*).

Paralogs are homologous genes that originate from the intragenomic duplication of an ancestral gene. Homologs that play a compensatory role can sometimes show similar phenotypes to each other when one of them is deficient (*Peng, 2019*), whereas other homologs might only offer partial compensation when the primary gene is more widely expressed or has higher activity levels. We

**Table 2.** Naa10, Naa11, and Naa12 peptides identified by LC-MS/MS analysis in Naa15 IP samples from WT and *Naa10*-KO mouse.

| | | Log2 LFQ intensity Naa15-IP | |
|---|---|---|---|
| Gene name | Peptide sequence | WT mouse | *Naa10*-KO mouse |
| *Naa12* | AMIENFSAK | 23.8144 | 27.5563 |
| *Naa12* | DLSQMADELRR | 25.2637 | 28.38 |
| *Naa12* | ENQGSTLPGSEEASQQENLAGGDSGSDGK | 21.299 | 22.09 |
| *Naa12* | RDLSQMADELR | - | 22.20 |
| *Naa12* | RDLSQMADELRR | - | 27.77 |
| *Naa10* | AALHLYSNTLNFQISEVEPK | 26.7672 | - |
| *Naa10* | AMIENFNAK | 27.3981 | - |
| *Naa10* | DLTQMADELRR | 25.5107 | - |
| *Naa10* | GNVLLSSGEACREEK | 25.0717 | - |
| *Naa10* | HMVLAALENK | 25.5293 | - |
| *Naa10* | NARPEDLMNMQHCNLLCLPENYQMK | 25.8928 | - |
| *Naa10* | YYFYHGLSWPQLSYIAEDENGK | 26.5915 | - |
| *Naa12;Naa11* | AALHLYSNTLNFQVSEVEPK | - | 27.3833 |
| *Naa12;Naa11* | YYFYHGLSWPQLSYIAEDEDGKIVGYVLAK | - | 25.2517 |
| *Naa12;Naa11;Naa10* | IVGYVLAK | 28.0873 | 25.7753 |
| *Naa12;Naa11;Naa10* | MEEDPDDVPHGHITSLAVK | 29.1069 | 29.265 |
| *Naa12;Naa11;Naa10* | MEEDPDDVPHGHITSLAVKR | 24.7605 | 21.7784 |
| *Naa12;Naa11;Naa10* | YVSLHVR | 22.8611 | 23.7383 |
| *Naa12;Naa11;Naa10* | YYADGEDAYAMK | - | 27.2083 |
| *Naa12;Naa11;Naa10* | YYADGEDAYAMKR | 27.2319 | 27.1689 |

Samples were run in technical duplicates and the average log2 LFQ intensity of the peptides is presented.

IP: immunoprecipitation; WT: wildtype; KO: knockout; LFQ: label-free quantification.

analyzed *Naa12* KO mice to see if they produced similar developmental defects to those in *Naa10* KO mice. KO mice for this gene were viable (*Supplementary file 1e*). Although there was initially a question of decreased fertility for the male mice, larger numbers of matings and litters did not bear this out (*Supplementary file 1d*), and necropsy and inspection of testes and seminal vesicles under a stereomicroscope did not reveal any macroscopic differences. Furthermore, the phenotypes (piebaldism and bilateral supernumerary ribs, *Figure 2B*) observed in *Naa10* KO mice with complete penetrance were not present in *Naa12* KO mice (*Figure 4—figure supplement 2B*). Overall, there were not any obvious phenotypes in these mice.

Matings between *Naa10*$^{+/-}$ *Naa12*$^{+/+}$ female mice and either *Naa10*$^{+/y}$ *Naa12*$^{+/-}$ or *Naa10*$^{+/y}$ *Naa12*$^{-/-}$ males produced zero male *Naa10*$^{-/y}$ *Naa12*$^{+/-}$ progeny, while also suggesting that compound heterozygous (*Naa10*$^{+/-}$ *Naa12*) female mice are produced at a rate much less than predicted by Mendelian ratios (*Supplementary file 1f, g*). Matings between surviving compound heterozygous (*Naa10*$^{+/-}$ *Naa12*$^{+/-}$) females and *Naa10*$^{+/Y}$ *Naa12*$^{+/-}$ males demonstrate that no live births occurred for *Naa10 Naa12* double-knockout (DKO) males (*Naa10*$^{-/Y}$ *Naa12*$^{-/-}$) (*Figure 6*). In addition, the average litter size was small when compared to the control (WT × WT) matings, suggesting embryonic lethality (*Table 3*). In order to determine whether lethality occurs during the embryonic stage, we genotyped E18.5 litters – just before birth. Consistent with our previous observations, we could not obtain any *Naa10*$^{-/Y}$ *Naa12*$^{-/-}$ embryos, and many embryos could not be genotyped because they were already in the midst of resorption (n = 23) (*Figure 6*). We checked an even earlier stage at E10.5 and also found zero *Naa10*$^{-/Y}$ *Naa12*$^{-/-}$ embryos, and also with far fewer resorptions at this stage (N = 3). Interestingly, we did observe *Naa10*$^{-/Y}$ *Naa12*$^{+/-}$ embryos where two of them displayed delayed developmental stage (appearing younger than E10.5) and another two embryos were lysed and had already begun degenerating (but despite this, we could at least genotype these embryos). This helps explain why only one *Naa10*$^{-/Y}$ *Naa12*$^{+/-}$ embryo was observed at E18.5.

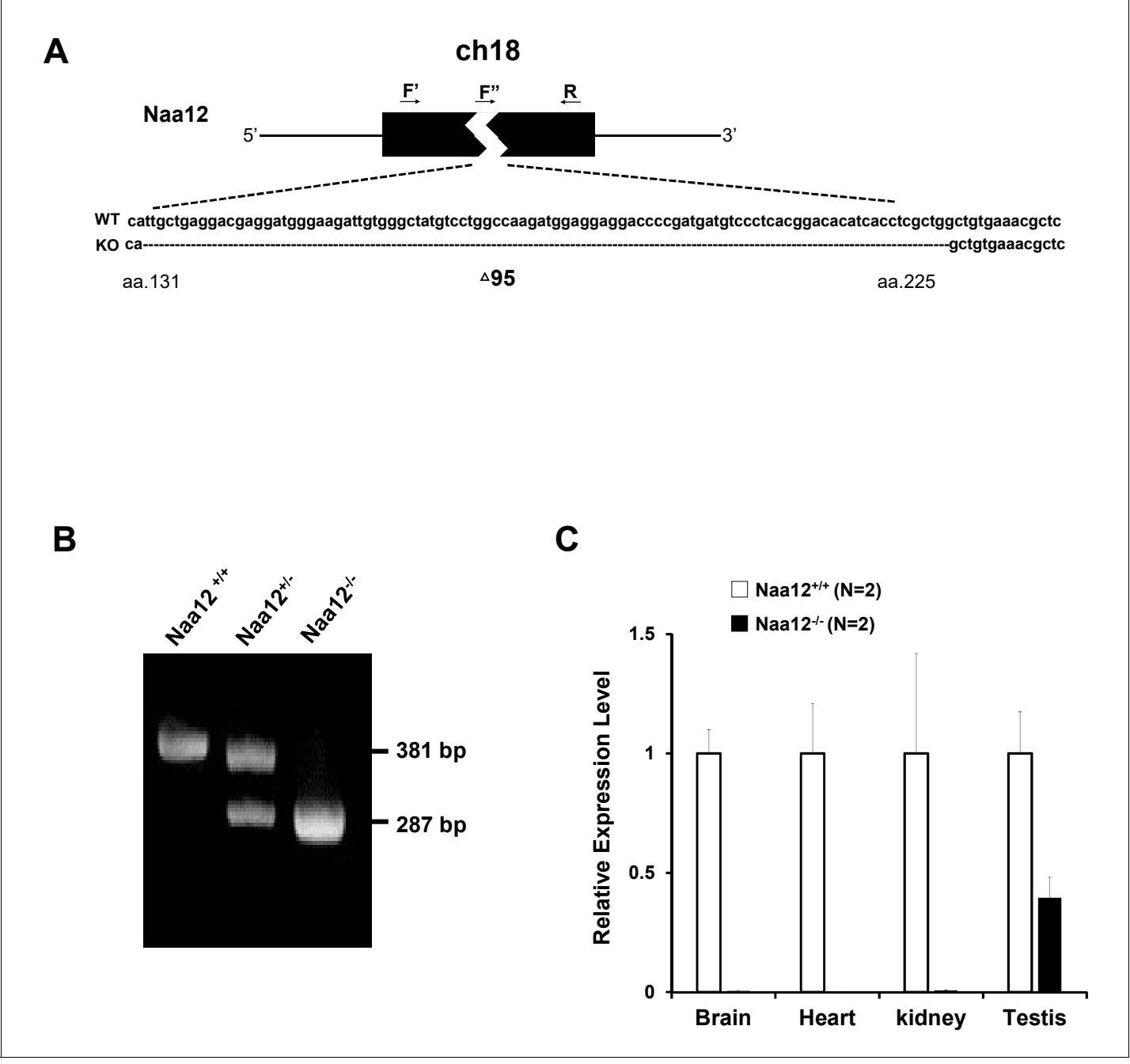

**Figure 5.** Generation of *Naa12* knockout (KO) mice. (**A**) Scheme of *Naa12* (Gm16286, UniProt: Q9CQX6) deletion used to generate *Naa12* KO mouse. 95 base pairs (131–225) were deleted. F': genomic DNA forward primer; F'': cDNA forward primer; R: reverse primer. (**B**) Genotyping of *Naa12* KO mice by PCR. Wildtype (WT) allele size was 381 bp and targeted allele size was 287 bp. (**C**) mRNA level of *Naa12* was analyzed in selected tissues by qPCR. Relative expression level of WT (white bars) and *Naa12* KO (black bars) after normalizing to that of GAPDH.

The online version of this article includes the following source data for figure 5:

**Source data 1.** Generation of *Naa12*KO mice.

Furthermore, *Naa10*[+/-] *Naa12*[-/-] female embryos were also lysed/degenerating at E10.5 and were not observed from that day onward. Matings between compound heterozygous females and *Naa10*[+/Y] *Naa12*[-/-] males also did not yield any *Naa10*[-/Y] *Naa12*[-/-] male mice at any embryonic stage examined, and only a couple of *Naa10*[+/-] *Naa12*[-/-] female mice at early stages of development (*Figure 6—figure supplement 1*), and the litter sizes were even smaller, suggesting increased

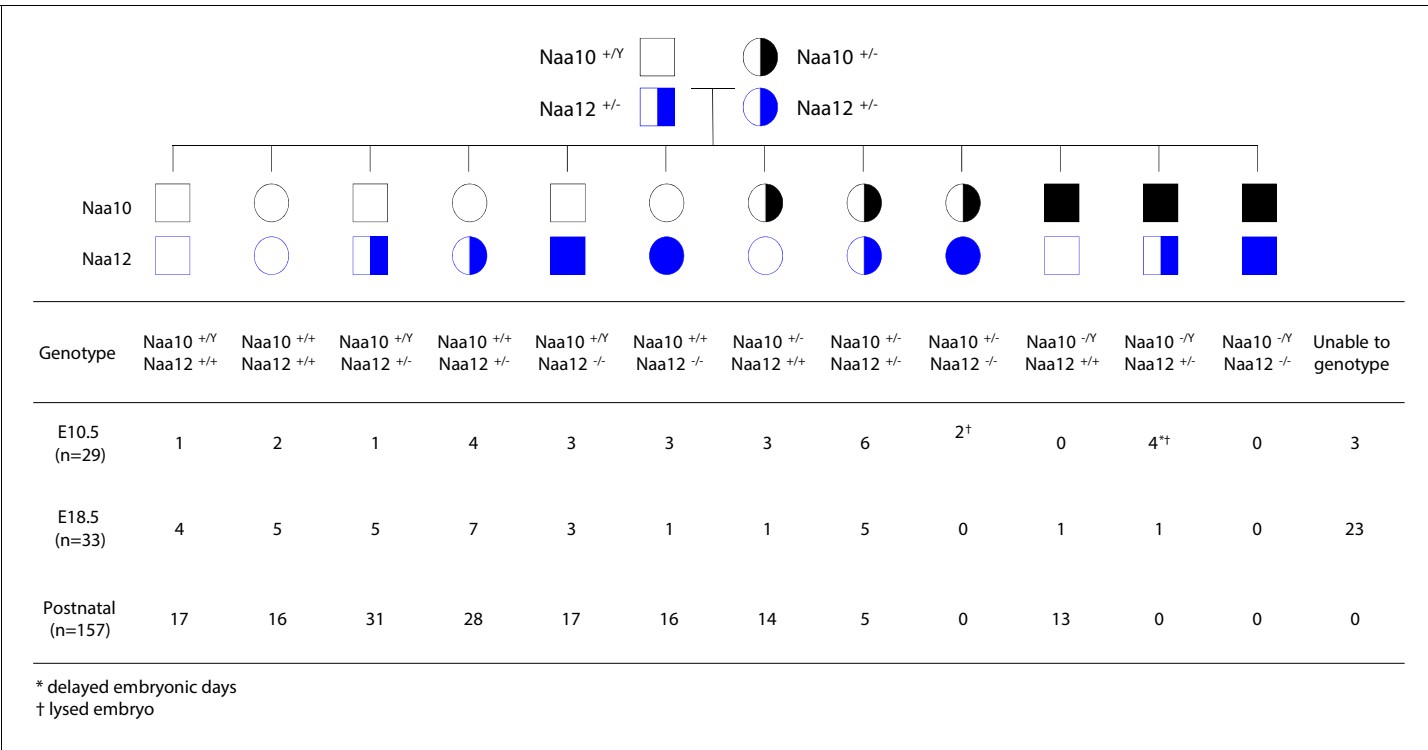

**Figure 6.** Lethality in *Naa10 Naa12* double-knockout (DKO) mice. *Naa10 Naa12* DKO exhibit embryonic lethality. Pedigree and genotypes of pups and embryos at E10.5 and E18.5 from *Naa10+/- Naa12+/-* female mice crossed to the *Naa10+/Y Naa12+/-* male mice.

The online version of this article includes the following figure supplement(s) for figure 6:

**Figure supplement 1.** Genotypes of offspring from *Naa10+/- Naa12+/-* female mice crossed to the *Naa10+/Y Naa12-/-* male mice.

**Figure supplement 2.** Comparisons of Mendelian predicted, observed, and model $D_4$ predicted offspring numbers for female genotypes (#1–#6) at each age.

**Figure supplement 3.** Comparisons of Mendelian predicted, observed, and model $D_4$ predicted offspring numbers for male genotypes (#7–#12) at each age.

**Figure supplement 4.** Comparisons of cumulative Mendelian predicted, observed, and model $D_4$ predicted offspring numbers for female genotypes (#1–#6) at each age.

**Figure supplement 5.** Comparisons of cumulative Mendelian predicted, observed, and model $D_4$ predicted offspring numbers for male genotypes (#7–#12) at each age.

embryonic lethality (*Table 3*). Consistent with this, we noted many resorptions at E12.5 and E18.5 that could not be genotyped. The number of living postnatal compound heterozygous female mice was also considerably lower than the predicted Mendelian ratios (*Figure 6*, *Figure 6—figure supplement 1*) and the surviving *Naa10+/- Naa12+/-* females were smaller in size than littermate controls (*Figure 7D*).

Due to the severe embryonic lethality observed in the *Naa10 Naa12* DKO male mice and the *Naa10+/- Naa12-/-* female mice, which was not seen in each single KO (*Naa10* KO or *Naa12* KO), it

**Table 3.** Litter size of Naa10 × Naa12 matings.

| Genotypes of Naa10; Naa12 breeders (♀ x ♂) | Total number of pups | Total number of litters | Average litter size (pups/litters) | SD of litter size |
|---|---|---|---|---|
| *Naa10+/+ Naa12+/+ × Naa10+/Y Naa12+/+* | 206 | 24 | 8.6 | 1.6 |
| *Naa10+/- Naa12+/- × Naa10+/Y Naa12+/-* | 157 | 32 | 4.9 | 1.5 |
| *Naa10+/- Naa12+/- × Naa10+/Y Naa12-/- ** | 225 | 63 | 3.6 | 1.7 |

*This mating was performed at IBR in Staten Island, New York, whereas the other two matings were performed at Ewha Womans University, Seoul, Republic of Korea.

SD: standard deviation.

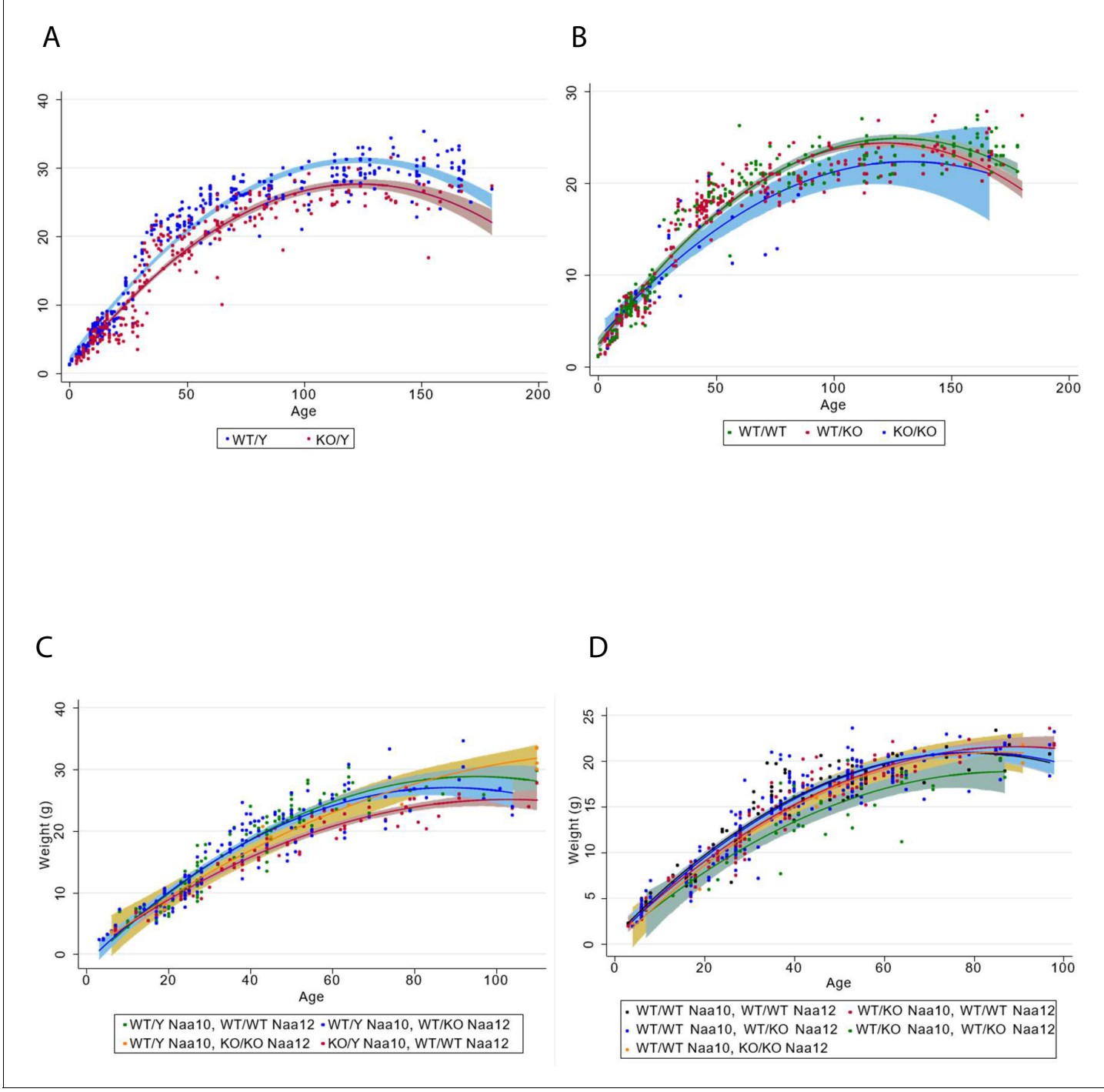

**Figure 7.** Decreased body weight in compound heterozygous females. (**A**) Male body weight for the *Naa10* mice on inbred genetic background (eight backcrosses to C57bl6/J). (**B**) Female body weight for the *Naa10* mice on inbred genetic background (eight backcrosses to C57bl6/J). (**C**) Male body weight for the *Naa10* and *Naa12* mice on mixed genetic background. (**D**) Female body weight for the *Naa10* and *Naa12* mice on mixed genetic background.

seems likely that, without compensation by Naa12, amino-terminal acetylation is disrupted in *Naa10 Naa12* DKO mice. Together, these data support the compensatory role of Naa12 in vivo.

## Genotype distribution modeling of *Naa10*- and *Naa12*-deficient offspring

The discrepancies we noted between the observed offspring genotype distributions and the expected Mendelian frequencies prompted us to examine the results from four matings (*Supplementary file 1h–m*) with the goal of understanding the effects of combined *Naa10* and *Naa12* mutations on embryonic and postnatal mortality. We created mathematical models to predict the observed genotype distribution at each age based on successive incorporation of assumptions of the lethality of specific offspring genotypes. Embryonic genotype data was obtained from two matings for which embryonic genotype data were obtained (*Supplementary file 1h, i*). Those matings were (1) $Naa10^{+/Y}$; $Naa12^{+/-}$ males crossed with $Naa10^{+/-}$; $Naa12^{+/-}$ females (*Figure 6*, *Supplementary file 1h*) and (2) $Naa10^{+/Y}$; $Naa12^{-/-}$ males crossed with $Naa10^{+/-}$; $Naa12^{+/-}$ females (*Figure 6—figure supplement 1*, *Supplementary file 1i*). The genotype numbering shown in *Supplementary file 1h* was used throughout this analysis, and the corresponding genotypes for all other crosses are aligned to have the same numbers. Each model described below adjusted the expected observed genotype frequencies at each age to account for loss of embryos or pups due to the predicted lethal effects of one or more genotypes by the method described in Materials and methods. Three stages of models (B–D) were compared with the expected Mendelian distribution (model A).

Model B assumed that the double KO male genotype 12 ($Naa10^{-/Y}$; $Naa12^{-/-}$) is lethal from very early in development based on the observation that this genotype was not found in any embryos or pups out of 483 obtained genotypes from all litters. Specifically, 0 out of an expected 7.9 were detected at E10.5 or earlier, 0 out of an expected 14.5 were detected at E18.5 or earlier, and 0 out of an expected 46.7 were detected by P3. Thus, the survival for genotype 12 was 0% for all ages examined.

Model C was developed from model B in two stages by incorporating separately observations that the male genotype 11 ($Naa10^{-/Y}$; $Naa12^{+/-}$) and the female genotype 6 ($Naa10^{+/-}$; $Naa12^{-/-}$) were lethal during mid to late fetal development. Based on the Mendelian model, 5 of 9.8 (51%) expected genotype 11 were detected by E10.5 but only 1 of expected 8.6 (11.6%) were identified on E12.5 or E18.5 and none were detected at P3. The five embryos that were present at E10.5 were noted to be lysed and/or developmentally delayed; the single E18.5 genotype 11 embryo was not observed to be abnormal. Based on the Mendelian model for genotype 6, 1 of 2.6 expected E8.5 embryos and 3 of 5.3 expected E10.5 embryos were identified. All three E10.5 embryos were identified as lysed. Genotype 6 was not identified after age E10.5. Cumulatively, 4 of 7.9 expected embryos detected by E10.5 and 0 of 38.8 expected embryos/pups thereafter.

Model D incorporated the assumptions of models B and C and added adjustments to the survival rates of genotype 5 ($Naa10^{+/-}$; $Naa12^{+/-}$) and genotype 10 ($Naa10^{-/Y}$; $Naa12^{+/+}$) based on the observations that these genotypes were underrepresented at late fetal ages or early postpartum. Genotype 5 was overrepresented during embryogenesis (31 identified but only 18.4 expected for all embryonic ages) but was underrepresented at P3 (17 of 42 expected) based on the expected Mendelian frequencies. A better analysis was achieved by comparing the observed genotype frequencies with those predicted by model C because the expected distributions are significantly affected by the lethal effects of the three genotypes considered in that model. In that case, the genotype overrepresentation during embryogenesis is somewhat less (31 identified but only 22.1 expected) but the underrepresentation at P3 is significantly increased (17 identified of 62 expected, or 27%). Using a model ($D_3$) that incorporated adjusted survival rates for genotypes 12, 11, 6, and 5, we found that genotype 10 remained underrepresented in the observed postnatal offspring counts. The subsequent model ($D_4$) incorporated adjustments to genotype 10 survival rates at E18.5 and P3 to account for this and then was slightly refined by adjusting other genotype survival rates to maximize the fit of all genotypes at all ages. The survival values for model $D_4$ are shown in *Table 4*. A comparison of the observed offspring numbers with those predicted by the Mendelian distribution and model $D_4$ is shown in *Figure 6—figure supplements 2–5*.

The observations of reduced survival for selected *Naa10/Naa12* mutants suggests that *Naa10* is the more dominant function (e.g., is able to provide *Naa12* functions more successfully than *Naa12* can provide *Naa10* functions) but that two copies of *Naa10* are required to replace complete loss of *Naa12* in females, possibly due to X-linked inactivation of *Naa10* during development. The stochastic

**Table 4.** Model $D_4$ genotype survival by age.

| #* | Genotype | E8.5 (%) | E10.5 (%) | E12.5 (%) | E18.5 (%) | Postnatal (%) |
|---|---|---|---|---|---|---|
| 12 | $Naa10^{-/Y}$; $Naa12^{-/-}$ | 0 | 0 | 0 | 0 | 0 |
| 11 | $Naa10^{-/Y}$; $Naa12^{+/-}$ | 40 | 35 | 10 | 10 | 0 |
| 6 | $Naa10^{+/-}$; $Naa12^{-/-}$ | 40 | 33 | 0 | 0 | 0 |
| 5 | $Naa10^{+/-}$; $Naa12^{+/-}$ | 100 | 100 | 100 | 100 | 35 |
| 10 | $Naa10^{-/Y}$; $Naa12^{+/+}$ | 100 | 100 | 100 | 55 | 55 |
| | All Others | 100 | 100 | 100 | 100 | 100 |

*Genotype number according to **Supplementary file 1h**.

E: embryonic day.

nature of X-linked inactivation in time and space may make *Naa10* functionality somewhat unpredictable during development in a background having a mixture of *Naa10* and *Naa12* mutations.

## Statistical examination of weight data in *Naa10*- and *Naa12*-deficient mice

To determine whether Naa10 and Naa12 are essential for viability and development, we examined the survival, weights, and growth rates of 688 *Naa10* and *Naa12* KO and WT mice. The genotypes of mice examined are listed in *Supplementary file 1n*. To avoid potential survival biases, only weights taken during the first 180 days were included. Growth curves are shown in *Figure 7*. Age and age-squared (the quadratic term) are both entered in the analyses; the quadratic term shows the degree to which the effect of age itself changes with age.

*Supplementary file 1o* shows the results in which the weight of *Naa10* mice in grams is regressed upon age, Naa10 KO status, and their interaction. Unsurprisingly, age predicts weight for males and females strongly, with growth slowing with age (first column). Though a strong negative effect of the KO is seen in for both males and females (second column), when both age and KO status are modeled together (third column), this effect all but disappears in females. Moreover, in females there is no interaction of KO status with age (fourth column), suggesting that the *Naa10* KO status itself has no significant effect on the growth rate in females. For males, however, the main effect of the KO remains when age is included in the model (third column) and the interaction is significant (fourth column), indicating that the *Naa10* KO both reduces weight of males overall and lowers the rate of growth.

Results of analyses of mixed-genetic background Naa10/Naa12 mice are shown in *Supplementary file 1p*. Effects of age and KOs on weight comprise the upper portion of the table, while the lower portion shows their effect on the rate of weight gain. Among females, a significant reduction of weight (above, second column) and in the rate of growth (below, first column) is seen among mice heterozygous for the Naa10 KO. There were no homozygous Naa10 KO female mixed-breed mice available to analyze as the matings were not setup to yield any such mice (so breeding patterns, not mortality in utero, are the reason for this absence). No significant effect on growth rate is seen for heterozygous or homozygous Naa12 KO (above, third column) or for their interactions with age (below, second column), and only the effects of the heterozygous Naa10 KO and its interaction with age are seen in the full model (below, third column). Thus, the Naa12 KO, whether heterozygous or homozygous, does not appear to reduce the weight or growth rate of females, while a heterozygous Naa10 KO is sufficient to reduce both weight and growth rate. Interestingly, when modeled together, both the Naa10 and the *Naa12* KOs significantly reduced weight (above, fourth column) and the interaction of the Naa10 and the *Naa12* heterozygous KOs significantly reduced weight (above, fifth column). As no female mice were both KO for *Naa10* and homozygous KO for *Naa12*, the effect of the interaction of those two factors could not be determined. The triple interaction of heterozygous *Naa10* KO, *Naa12* KO, and age was weakly significant, suggesting that the presence of both KOs affects growth rate above and beyond the effects of each KO independently (below, fourth column). No males with KOs of both *Naa10* and *Naa12* were born, so no test of their interaction was possible. An effect was seen for the Naa10 KO on weight when modeled with age and age² (second column), and the significant interaction of the *Naa10* KO with age and age² (third

column) shows that the *Naa10* KO in males reduces the growth rate. As with females, no significant effect of a *Naa12 KO*, whether heterozygous or homozygous, was seen in males, nor is there a significant interaction with age (fourth column). When the interactions of age with both *Naa10* and *Naa12* KO status are entered in one model, *Naa10* alone is seen to reduce growth rates (fifth column).

## Discussion

We have shown that Naa10 deficiency results in pleotropic developmental defects in two different Naa10-deficient mouse models. Similar to infant mortality in some OS males, the lethality of *Naa10* KO mice increased dramatically in pups in the first three days of life (*Figure 1B*). Defects in kidney, brain, pigmentation (piebaldism), and ribs were observed during embryonic or early postnatal stages in some mice (*Figure 2B, C*). These observed phenotypes overlap with some of the phenotypes found in surviving humans with OS, including supernumerary vertebrae and hydrocephaly, although piebaldism has not been reported to date in any humans. However, the puzzling lack of embryonic lethality in the *Naa10* KO mice prompted us to discover *Naa12* as a possible compensatory NAT, with Naa10-like amino-terminal acetylation activity (*Figure 4A*), with an interaction between Naa15 and Naa12 (*Figure 4B*), and with enzymatic activity in a chimeric complex with human NAA15 (*Figure 4C*). In addition, co-immunoprecipitation of endogenous Naa15 from Naa10 KO mouse tissues followed by mass spectrometry analysis (*Table 2*) and amino-terminal acetylation assays (*Figure 3C*) fully supports that the endogenous Naa12-Naa15 complexes produces NatA activity. Finally, we found genetic proof of the compensatory activity of Naa12 in mice when we observed embryonic lethality in in *Naa10 Naa12* DKO male and *Naa10$^{+/-}$ Naa12$^{-/-}$* female mice (*Figure 6*). This compensation by Naa12 explains the mouse proteomics data indicating normal amino-terminal acetylation in Naa10 KO mice (*Figure 3A*). We have confirmed the expression of Naa12 in various tissues using qPCR (*Figure 4—figure supplement 1A*).

Gene duplication has long been believed to be a major driving force in evolution that provides genetic novelty in organisms. Paralogous genes, originating by small-scale or whole-genome duplication, overlap functional roles for each other and can completely or partially compensate for the loss of the duplicate gene (*Peng, 2019*; *Veitia, 2017*). There is not yet any human reported with complete KO for *NAA10*. There is one published truncating variant in the C-terminal portion of NAA10 in a male patient with microphthalmia (*Cheng et al., 2019*), but unfortunately there are no cell lines available from this family to confirm whether any truncated NAA10 protein is expressed, as was shown with a splice-site mutation in a Lenz microphthalmia family (*Esmailpour et al., 2014*). NAA10 was also identified in screens for essential genes in human cell lines (*Blomen et al., 2015*; *Wang et al., 2015*), so it seems unlikely that an unknown NAA10-like paralogous gene exists in humans, other than the already known NAA11.

The pleiotropic phenotypes shown in *Naa10* KO mice, including hypopigmentation and supernumerary ribs with a penetrance of 100%, were not observed in the *Naa12* KO mice. Naa10 itself has been described to have N-ε-acetyl-activity towards internal lysine residues of proteins involved in various disease- and development-related signaling pathways (*Lee et al., 2018*), although its acetylation of some substrates is controversial (*Magin et al., 2016*; *Vo, 2020*). Since the Nt-acetylome appears to be globally intact in MEFs from *Naa10* KO mice (*Figure 3A*), it is possible that the presented phenotypes could be due to the loss of Naa10-specific N-ε-acetyl-activity or non-catalytic roles of Naa10 (*Aksnes et al., 2019*). Alternatively, the quantitative expression of Naa10 and Naa12 might be different within or between tissues, which might then explain why there is clearly a phenotype for *Naa10$^{+/-}$ Naa12$^{+/-}$* female mice (not born at Mendelian ratios and the few that are born are usually much smaller) but no apparent phenotype in *Naa10$^{+/+}$ Naa12$^{-/-}$* female mice. It seems likely that the mechanism cannot be simply additive between two equally expressed proteins, because if the expression of each protein is theoretically set at an arbitrary unit of 10, then *Naa10$^{+/-}$ Naa12$^{+/-}$* female mice might possibly have half as much of each protein, so that the total dose of both proteins together would be 10, instead of 20. Likewise, the total dose of both proteins together would also be predicted to be 10 in a *Naa10$^{+/+}$ Naa12$^{-/-}$* female. Yet, the *Naa10$^{+/-}$ Naa12$^{+/-}$* female mice have a phenotype, whereas the *Naa10$^{+/+}$ Naa12$^{-/-}$* female mice do not (*Figure 6*). Therefore, other explanations could include different tissue-specific dosages of each protein, different expression between different tissues, possible X-chromosome skewing for the X-linked *Naa10* in different tissues, or different functions of the two enzymes, including Naa10-specific N-ε-acetyl-activity or non-catalytic

roles of Naa10 (*Aksnes et al., 2019*). These questions remain unanswered and are worth exploring in future studies. It is worth highlighting that X-chromosome inactivation could certainly be one explanation, given that males that are $Naa10^{+/Y}$ with $Naa12^{+/-}$ or $Naa12^{-/-}$ show expected survival rates, whereas females that are $Naa10^{+/-}$ with $Naa12^{+/-}$ show ~35% survival but with $Naa12^{-/-}$ show 0% survival (*Table 4*).

There are several clinical features that were presented in the original description of OS (*Rope et al., 2011*) which can now be better understood in light of the phenotypes found in the KO mouse model. For example, all of the affected children in the first families with OS were noted to have large and, in some cases, persistently open fontanels (*Myklebust et al., 2015*; *Rope et al., 2011*). For one child (family 1, individual II-1), CT scanning revealed cerebral atrophy with enlarged ventricles, and in another child (family 1, individual III-4), there was evidence on magnetic resonance imaging (MRI) of 'moderate lateral and third ventricular dilatation without identified cause.' Lastly, all of the children had respiratory depression and apneic episodes, along with varying course of hypotonia and/or hypertonia (including documented hyperreflexia in at least one case [family 2, individual III-2]). In retrospect, it seems that these clinical features could be consistent with mild hydrocephaly in these probands with OS, which resolved over time. This is also consistent with the ventriculomegaly reported in several female OS probands with missense mutations in *NAA10*, along with ventriculomegaly in one other male proband who died in the first week of life, with generalized hypotonia and lack of spontaneous respirations (*Saunier et al., 2016*). One of the female patients with an Arg83Cys mutation in Naa10 (#9 in Table 1 of that paper) was reported as having intraventricular hemorrhage in the occipital horn, hypoxic-ischemic encephalopathy, and a ventriculo-peritoneal shunt. It is possible that this sequence of events is compatible with hydrocephaly with clinical signs and symptoms that required the placement of the shunt.

There are additional cardiac and skeletal features that are also worth re-examining in light of these new findings. In some of the original cases of OS, there were varying levels of pulmonary valve stenosis detected on echocardiography, along with some documentation of pulmonary hypoplasia (*Rope et al., 2011*). For example, individual III-7 in family 1 was found on echocardiography to have small persistent ductus arteriosus, a mildly decreased left ventricular systolic function, an abnormal appearing aortic valve, an enlargement of the right ventricle, decreased right ventricular systolic function, and persistence of the foramen ovale. Individual III-6 from this same extended family was found on echocardiography to have a thickened bicuspid aortic valve and mild pulmonary hypertension. One of the OS female patients with an Arg83Cys mutation in *NAA10* was reported to have 'supernumerary vertebrae' (*Saunier et al., 2016*). Prompted by our findings of supernumerary ribs in the mice, we obtained an MRI report for this patient, in which the radiologist concluded that there appeared to be 25 distinct vertebrae, as opposed to the usual 24, with a suggestion of a 13th rib, at least on the right. The report went on to state that "the vertebrae represent seven cervical vertebrae, 13 rib-bearing thoracic vertebrae, and five lumbar vertebrae, and the L1 vertebra is mildly dysmorphic, with a suggestion of anterior breaking." In addition, chest and abdominal X-rays from two of the brothers in generation VI of a family with microphthalmia demonstrated the presence of 13 rib-bearing thoracic vertebrae, alongside the dramatic scoliosis in both individuals. Four other females carrying mutations in Naa10 were reported as having either pectus carinatum or excavatum (*Saunier et al., 2016*), one of the boys with OS (family 1, individual III-4) was noted to have pectus excavatum, and retrospective review of some of the clinical photographs appears to show mild pectus excavatum in individual III-6 of the same family. Studies of human populations have shown that the levels of transition may be shifted cephalad, resulting in 23 mobile vertebrae, or shifted caudad, resulting in 25 presacral vertebrae. Such variations may occur in 2–11% of the population (*Bornstein and Peterson, 1966*). In addition, the number of ribs can also vary in mice as a result of teratogenic and genetic influences (*Mclaren and Michie, 1958*; *Chernoff and Rogers, 2004*). However, the complete penetrance for supernumerary ribs in the *Naa10*-deficient mice, along with the presence of extras ribs in some of the patients, suggests that there is a pathway common to humans and mice that is altered by mutations involving *NAA10*.

Several mouse mutants show similar cardiac or skeletal phenotypes to the *Naa10*-deficient mice. *Pax3* mutants phenocopy our $Naa10^{-/-}$ mutants as $Pax3^{+/-}$ adults exhibit 100% piebaldism and exhibit neural crest (NC)-related PTA/DORV with concomitant VSDs (*Conway et al., 1997a*; *Conway et al., 1997b*; *van den Hoff and Moorman, 2000*; *Olaopa et al., 2011*). *Pax3* systemic nulls also have skeletal defects due to abnormal somite morphogenesis (*Henderson et al., 1999*;

*Dickman et al., 1999*). Moreover, *Pax3* cKOs demonstrated that NC-specific deletion is sufficient to cause DORV/VSDs and death at birth (*Olaopa et al., 2011*; *Koushik et al., 2002*), and that restricted deletion within the neuroepithelium causes congenital hydrocephalus (*Zhou and Conway, 2016*). While *Pax7* systemic deletion does not cause NC-associated defects, it does exhibit overlapping expression, and Pax3-Pax7 compound heterozygous mice develop hydrocephalus (*Zhou and Conway, 2016*), suggesting combinatorial function. *Hox C8$^{-/-}$* mice exhibit an extra rib and an extra rib articulating with the sternum (*Le Mouellic et al., 1992*; *Juan and Ruddle, 2003*), and an unfused sacral vertebra, which lead to 27 presacral vertebrae (*van den Akker et al., 2001*), as seen in our model. *Hox A4$^{-/-}$* mice described in *Horan et al., 1994* show cervical fusions of C2/C3, a rib on C7 not fully penetrant and sternal defects with bone ossification anomalies. *Hox A5$^{-/-}$* mice display numerous cervico-thoracic defects such as a rib process coming from the seventh cervical vertebra, an increase in the number of sternebrae and total number of ribs (*Jeannotte et al., 1993*). Both *Hox A4$^{-/-}$* and *A5$^{-/-}$* mice exhibit an extra rib articulating with the sternum. *Hox D3$^{-/-}$* mice are the only Hox gene mutation leading to cervical fusion of both the atlas and axis (*Condie and Capecchi, 1993*). *Hox A9$^{-/-}$* mice have anteriorization of both sacral and lumbar parts, with an extra pair of ribs at the lumbar level. *Hox A9$^{-/-}$* mice do not have any relevant sternal defect (*Fromental-Ramain et al., 1996*). *Hox B9$^{-/-}$* mice have an extra rib articulating with the sternum and 14 pairs of rib (*Chen and Capecchi, 1997*). These phenotypes, especially *Hox C8*, share common features with the *Naa10*-deficient mice. This phenotype is also close to the *Rpl38$^{-/-}$* phenotype (*Kondrashov et al., 2011*), except for the sacral fusion described in *Rpl38$^{-/-}$* mice. Interestingly, it was shown that *Hox* genes were dysregulated in this genotype. The skeletal findings and comparison to other mutant mice suggest a pattern consistent with a homeotic anterior transformation hypothesis.

The developmental role of Naa10 in mice has been previously described (*Lee et al., 2017*). Lee et al. reported embryonic lethality at E12.5–14.5 and beyond (due to placental defects), hydrocephaly, postnatal growth retardation, and maternal effect lethality in *Naa10* KO mice and suggested that genomic imprinting dysregulation is associated with those developmental phenotypes. In the present study, hydrocephaly and postnatal growth retardation were also apparent, but embryonic lethality was not observed, which prompted the search for and discovery of Naa12. The previous paper (*Lee et al., 2017*) did not report the piebaldism, homeotic anterior transformation, hydronephrosis, and genital defects (such as seminal vesicle malformation and hydrometrocolpos), nor did it explain the cause of death in the first day of life, which is at least partly due to congenital heart defects, as reported herein. A more recent paper from the same group reported that conventional and adipose-specific Naa10p deletions in mice resulted in increased energy expenditure, thermogenesis, and beige adipocyte differentiation in the surviving mice (*Lee et al., 2019*), although the authors do not comment on whether any of the male mice used in that study starting at age 5 weeks ended up developing hydrocephaly and/or hydronephrosis, which we have observed in older mice. Although the *Lee et al., 2017* paper reported a very high maternal effect lethality rate of 90% (46/51) (otherwise stated as a survival rate of 10% [5/51]) for newborns in matings following *Naa10$^{-/-}$* female and C57BL/6J WT male intercrossing, this rate was only 29% (37/127) in this same mating herein in the first 24 hr of life and with a total death rate in the first three days for all newborns of 46/127, or 36% (*Supplementary file 1d*), with this result deriving from a larger number of mating pairs, litters, and pups. Although this rate of 36% is higher than that seen with matings involving *Naa10$^{+/-}$* females (15.8% and 13.6%) (*Supplementary file 1d*), the explanation for this ~20% difference in survival in the first three days of life could involve differences in maternal care provided by the *Naa10$^{+/-}$* and *Naa10$^{-/-}$* females, but this would have to be investigated in future studies, involving detailed behavioral and cognitive assessment of the dams.

The reasons for the differences between the studies in regards to maternal effect lethality and in utero lethality are unknown at present. Whilst Lee et al. deleted *Naa10* exons 2–6 (*Lee et al., 2017*), the current study deleted *Naa10* exons 1–4 *or used an allele Naa10$^{tm1a}$* expressing β-galactosidase instead of the *Naa10* gene (*Figure 1—figure supplement 1D*), and there was not any significant embryonic lethality in either line (*Supplementary file 1a, b*). All three of these mouse models were made using 129Sv/Ev ES cells, and all three are nulls lacking Naa10 protein. It is the case that the previous study used the Cre/loxP system to generate the *Naa10* KO mice, where a floxed Naa10 female mouse was crossed with the Ella-Cre transgenic male mouse expressing Cre recombinase for germ line deletion of loxP-flanked *Naa10*, whereas our mice were made using standard gene-

targeting methods without the use of Cre recombinase, but it is not clear how this would have resulted in embryonic lethality, particularly as these mice were only used after 'at least six generations of backcross with C57BL/6 mice,' which are noted by the authors to be the substrain C57BL/6JNarl, first established at the Animal Center of National Research Institute from the Jackson Laboratory (JAX) in 1995. The explanation for differences in embryonic lethality might be more likely due to different combinations of modifying alleles that are present in the different C57BL/6J substrain genetic backgrounds, rather than differences in our model systems, and future plans will address this after back-crossing more than 20 generations to C57BL/6J (imported annually from JAX) to achieve an entirely inbred line. The impact of genetic background is supported by the observation that additional null alleles on mixed genetic backgrounds, made during the process of generating missense mouse models for OS, have far less penetrance for a range of the various phenotypes, including much less perinatal lethality (unpublished observations).

In conclusion, our study provides strong evidence that Naa10, the catalytic subunit of NatA, is critical for normal development in mice. Furthermore, this study explains the puzzle regarding the lack of complete embryonic lethality in the *Naa10* KO mice due to the discovery of a second mouse *Naa10* paralog, which, unlike *Naa11*, is expressed in the heart as well as other tissues. Taken together, our findings suggest that the newly identified Naa12 can functionally rescue Naa10 loss and act as a catalytic subunit in mouse NatA complexes.

# Materials and methods

## Key resources table

| Reagent type (species) or resource | Designation | Source or reference | Identifiers | Additional information |
|---|---|---|---|---|
| Gene (*Mus musculus*) | Naa10 | GenBank | MGI:MGI:1915255 | |
| Gene (*M. musculus*) | Naa15 | GenBank | MGI:MGI:1922088 | |
| Gene (*M. musculus*) | Naa11 | GenBank | MGI:MGI:2141314 | |
| Gene (*M. musculus*) | Naa12 | This paper | Gm16286, UniProt: Q9CQX6 | Provided by corresponding author, Gholson J. Lyon |
| Genetic reagent (*M. musculus*) | Naa10$^{-/-}$ | Nature Communication **Yoon et al., 2014** | | Provided by corresponding author, Goo Taeg Oh |
| Genetic reagent (*M. musculus*) | Naa12-/- | This paper | Gm16286, UniProt: Q9CQX6 | Provided by corresponding author, Gholson J. Lyon |
| Cell line (*Homo sapiens*) | HEK293 (normal, embryonic kidney cells) | ATCC | CRL-1573 | |
| Biological sample (*M. musculus*) | Primary mouse embryonic fibroblasts | This paper | | Freshly isolated from mouse embryos (E13.5) |
| Antibody | Anti-Naa10 (rabbit polyclonal) | Abcam | Cat# ab155687 | (1:1000) |
| Antibody | Anti-Naa10 (rabbit polyclonal) | Protein Tech | Cat# 14803-1-AP | (1:3000) |
| Antibody | Anti-Naa10 (rabbit monoclonal) | Cell Signaling | Cat# 13357 | (1:1000) |

*Continued on next page*

*Continued*

| Reagent type (species) or resource | Designation | Source or reference | Identifiers | Additional information |
|---|---|---|---|---|
| Antibody | Anti-Naa10 (goat polyclonal) | Santa Cruz | Cat# sc-33256 | (1:1000) |
| Antibody | Anti-Naa10 (rabbit polyclonal) | Santa Cruz | Cat# sc-33820 | (1:1000) |
| Antibody | Anti-Naa11 (rabbit polyclonal) | Novus Biologicals | Cat# NBP1-90853 | (1:1000) |
| Antibody | Anti-Naa15/NARG1 (mouse monoclonal) | Abcam | Cat# ab60065 | (1:1000) |
| Antibody | Anti-NAA15 (rabbit polyclonal) | Biochemical Journal (reference 12 in this paper) *Arnesen et al., 2005* | | (1:2000) Provided by author Thomas Arnesen, |
| Antibody | Anti-NAA50 (rabbit polyclonal) | LifeSpan BioSciences | Cat# LS-C81324-100 | (1:3000) |
| Antibody | Anti-FLAG (rabbit polyclonal) | Sigma-Aldrich | Cat# F7425 | (2 µg/mL) |
| Antibody | Anti-GAPDH (mouse monoclonal) | Abcam | Cat# ab9484 | (1:3000) |
| Antibody | Anti-actin (goat polyclonal) | Santa Cruz | Cat# 1615 | (1:3000) |
| Antibody | Anti-GST (mouse monoclonal) | GenScript | Cat# A00865 | (1 µg/mL) |
| Antibody | Anti-V5 (mouse monoclonal) | Life Technologies | Cat# R960-25 | (1:1000) |
| Antibody | Anti-Naa12 (rabbit polyclonal) | This paper | Gm16286, UniProt: Q9CQX6 | C-terminus (aa191-205: QENLAGGDS GSDGKD-C) conjugated to OVA by PrimmBiotech Provided by corresponding author, Gholson J. Lyon |
| Sequence-based reagent | mNaa10-Exon2/3_F | This paper | PCR primers | ctcttggccccagctttctt Provided by corresponding author, Goo Taeg Oh |
| Sequence-based reagent | mNaa10-Exon3/4_R | This paper | PCR primers | tcgtctgggtcctcttccat Provided by corresponding author, Goo Taeg Oh |
| Sequence-based reagent | mNaa11_F | This paper | PCR primers | accccacaagcaaagacagtg Provided by corresponding author, Goo Taeg Oh |

*Continued on next page*

*Continued*

| Reagent type (species) or resource | Designation | Source or reference | Identifiers | Additional information |
|---|---|---|---|---|
| Sequence-based reagent | mNaa11_R | This paper | PCR primers | agcgatgctcaggaaatgctct Provided by corresponding author, Goo Taeg Oh |
| Sequence-based reagent | mNaa12 (Gm16286)_F | This paper | PCR primers | acgcgtatgctatgaagcga Provided by corresponding author, Gholson J. Lyon |
| Sequence-based reagent | mNaa12 (Gm16286)__R | This paper | PCR primers | ccaggaagtgtgctaccctg Provided by corresponding author, Gholson J. Lyon |
| Sequence-based reagent | mNaa15_F | This paper | PCR primers | gcagagcatgg agaaaccct Provided by corresponding author, Gholson J. Lyon |
| Sequence-based reagent | mNaa15_R | This paper | PCR primers | tctcaaacctctgcgaacca Provided by corresponding author, Gholson J. Lyon |
| Sequence-based reagent | mNaa50_F | This paper | PCR primers | taggatgccttgcaccttacc Provided by corresponding author, Gholson J. Lyon |
| Sequence-based reagent | mNaa50_R | This paper | PCR primers | gtcaatcgctgactcattgct Provided by corresponding author, Gholson J. Lyon |
| Sequence-based reagent | mGAPDH_F | This paper | PCR primers | aggtcggtgtgaacggatttg Provided by corresponding author, Gholson J. Lyon |
| Sequence-based reagent | mGAPDH_R | This paper | PCR primers | tgtagaccatgtagtt gaggtca Provided by corresponding author, Gholson J. Lyon |
| Sequence-based reagent | mACTB_F | This paper | PCR primers | ggctgtattcccctccatcg Provided by corresponding author, Gholson J. Lyon |
| Sequence-based reagent | mACTB_R | This paper | PCR primers | ccagttggtaac aatgccatgt Provided by corresponding author, Gholson J. Lyon |
| Software, algorithm | Zen 3.0 SR | ZEISS | Version 16.0.1.306 | Black 64bit edition |
| Other | Alcian Blue 8GX | Sigma-Aldrich | Cat# A5268 | 0.03% |
| Other | Alizarin Red | Sigma-Aldrich | Cat# A5533 | 0.05% |
| Other | Hematoxylin | Sigma-Aldrich | Cat# MHS80 | |
| Other | Eosin | Sigma-Aldrich | Cat# HT110116 | |

## Mice

All experiments were performed in accordance with guidelines of International Animal Care and Use Committee (IACUC) of Ewha Womans University (protocol #18-012), Cold Spring Harbor Laboratory (CSHL) protocol #579961-18, and Institute for Basic Research in Developmental Disabilities (IBR) (protocol #456). At CSHL and IBR, any matings that required genotyping were screened on a daily basis by animal husbandry staff, with notation of how many newborn pups were present each morning, but with paw tattoo and tail genotyping not being performed until day 3 of life, so as to not disturb the litters and thus to not increase the risk for maternal rejection of the litter. The stock of C57BL/6J was replenished annually from Jackson Laboratory so as to avoid genetic drift from the JAX inbred line.

## Generation of *Naa10*-deficient mice

The *Naa10* KO mice were generated as previously described (*Yoon et al., 2014*). *Naa10*$^{tm1a}$ [B6;129P2-Ard1$^{tm1a(Eucomm)Gto}$/J] (*Naa10*$^{tm1a}$) mice, used for *Naa10* reporter mouse, were generated using standard method based on a standard gene-targeting in E14 embryonic stem (ES) cells (129/Sv) by using a targeting vector from EUCOMM. Correctly targeted ES clones were used for blastocyst microinjection and generation of chimeric mice. Chimeric mice were crossed to C57BL/6J mice, and then the progeny were backcrossed to C57BL/6J for more than 10 generations. The *Naa10*-deficient mice used in the weight analyses were derived from mice backcrossed eight times to a C57BL/6J inbred genetic background, and this was confirmed with genome scanning at the Jackson Laboratory, showing heterozygosity for only one marker for 129S1/SvImJ out of 290 autosomal markers tested, thus giving a percentage of C57BL/6J of 99.66%.

## Generation of *Naa12* (Gm16286, UniProt: Q9CQX6) KO mice

The mice were made using standard methods by microinjection of CRISPR reagent mix into zygotes obtained from the mating of B6D2F1 females (i.e., 50% C57BL/6J, 50% DBA/2J [D2]) females to inbred C57BL/6J males. The guide RNA was produced and validated from Sigma using a Cel1-nuclease assay, and the most active guide was selected, which was Naa12_0_125 (C9587), with a target sequence of GAGCGTTTCACAGCCAGCG and including the targeting cr-RNA sequence and the tracrRNA portion. The indels were transmitted by breeding again to inbred C57BL/6J males, and the resulting progeny were interbred on a mixed genetic background of approximately 12.5% DBA/2J (D2)/87.5% C57BL/6J, for use in the reported experiments, including the weight analyses. Progeny from these mice have been backcrossed to C57BL/6J for more than 10 generations, with no discernible new phenotypes emerging. Genomic DNA was isolated from paw and tail. DNA was screened for mutations using PCR and Surveyor assay (*Qiu et al., 2004*), followed by Sanger sequencing of selected clones and the use of CRISP-ID (*Dehairs et al., 2016*) to identify putative deletions.

## Primers for mice genotyping

The primers used for *Naa10 KO* and *Naa10*$^{tm1a}$ genotyping were Naa10-F: 5′-cctcacgtaatgctctgcaa-3′, Naa10-neo-F: 5′-acgcgtcaccttaat-atgcg-3′, Naa10-R: 5′-tgaaagttgagggtgttgga-3′, Naa10$^{tm1a}$-F: 5′-gcacactctctgaattggac-3′, Naa10$^{tm1a}$-neo-F: 5′-ggccgcttttctggattcat-3′, and Naa10$^{tm1a}$-R: 5′-gcaggggaataaggcattgg-3′. The primers used for Naa12 KO were Naa12 Surveyor F: 5′-gctccacctcgc-taacctgg-3′, Naa12 Surveyor R: 5′-gccagatgacctgatgaacatgc-3′ and HEX-Naa12 Surveyor F: 5′-gctccacctcgctaacctgg-3′.

## Antibodies

The following antibodies were used: rabbit anti-Naa10 (Abcam #ab155687), rabbit anti-Naa10 (Protein Tech #14803-1-AP), rabbit monoclonal anti-NAA10 (Cell Signaling, #13357), goat anti-Naa10 (Santa Cruz, #sc-33256), rabbit anti-Naa10 (Santa Cruz, #sc-33820), rabbit anti-Naa11 (Novus Biologicals; #NBP1-90853), mouse anti-Naa15/NARG1 (Abcam; #ab60065), rabbit polyclonal anti-NAA15 (*Arnesen et al., 2005*), rabbit anti-Naa50 (LifeSpan BioSciences; #LS-C81324-100), rabbit anti-FLAG (Sigma; #F7425), mouse anti-GAPDH (Abcam; #ab9484), goat anti-actin (Santa Cruz, #1615), mouse anti-GST (GenScript; #A00865), and mouse anti-V5 (Life Technologies; #R960-25). The antibody

against the potential mNaa10 paralog mNaa12 (Gm16286, UniProt: Q9CQX6) was raised in rabbits after immunization with a synthetic peptide of the Naa12 C-terminus (aa191-205: QENLAGGDSG SDGKD-C) conjugated to OVA by PrimmBiotech.

## Alcian Blue and Alizarin Red co-staining of skeletons

After the skin and internal organs were removed, embryos were fixed in 95% ethanol (EtOH) for 4 hr, then in 100% acetone for overnight. Embryos were stained with 0.03% Alcian Blue 8GX in ethanol/acetic acid (4:1 v/v) for overnight and kept in 1% KOH for 2 days until they became clearly visible, followed by staining with 0.05% Alizarin Red in 1% KOH for 4 hr. After washing with 100% glycerol/1% KOH (1:1 v/v), skeletons were kept in 100% glycerol.

## Isolation and imaging of mouse embryos

Timed matings were performed either by using the presence of a vaginal plug to assess fertilization. The morning vaginal plug was designated E0.5. Pregnant mice were sacrificed at several time points after conception. The embryos were isolated in ice-cold PBS with 1% FBS and washed three times in ice-cold PBS. Embryos were imaged using a Zeiss Axiozoom V16 with Zen software and merged 50 slides between Z-stack intervals.

## β-Galactosidase staining

Isolated E10.5 embryos were incubated in fixation solution (4% paraformaldehyde) at 4°C for 25 min. Samples were washed in ice-cold PBS and then incubated in permeabilization solution (PBS containing 0.01% Na deoxycholate, 0.02% Nonidet-P40, 2 mM $MgCl_2$) for 20 min at 4°C. Subsequently, samples were incubated in β-gal staining solution (PBS containing 1 mg/mL X-Gal, 5 mM potassium ferrocyanide, 5 mM potassium ferricyanide, 0.02% Nonidet-P40, 2 mM $MgCl_2$) at 37°C overnight. Following β-gal staining, samples were washed with PBS and incubated in fixation solution at 4°C for storage.

## Hematoxylin and eosin staining

Isolated kidney tissues at E18.5 and P3 were fixed with 4% paraformaldehyde at 4°C for overnight and embedded in paraffin. Samples were sectioned at 8 μm thick and stained with hematoxylin (MHS80, Sigma) and eosin (HT110116, Sigma) for morphology.

## Cloning

Full-length mouse Naa10 and Naa12 (Gm16286, UniProt: Q9CQX6) expression vectors were separately constructed using a pMAL-c5x vector. In both cases, the catalytic subunit contained an N-terminal uncleavable MBP-tag. Bacterial expression vectors of mNATs were cloned from cDNA generated from mouse liver or testes. mRNA was isolated using the Oligotex direct mRNA kit (Qiagen) according to the manufacturer's recommendations. 1 μg RNA was reverse transcribed with Superscript IV reverse transcriptase (Thermo Fisher) and Oligo dT(18) primer. The PCR product was digested and cloned into BamHI restriction sites of pGEX-4T1 (GE Healthcare), pMAL-p5X and p3xFLAG-CMV10 (Sigma-Aldrich) using standard techniques. All constructs were sequenced to validate correct insert and orientation.

## Primers for cloning

cDNA was amplified using the primers CCG GGA TCC ATG AAC ATC CGC AAT and CTG GGA TCC CTA GGA GGC AGA GTC AGA for mNaa10 variants, CCG GGA TCC ATG AAC ATC CGC AA T GC and CTG GGA TCC CTA GGA GAT GGA TCC AA GTC for mNaa11, CCG GGA TCC ATG AAC ATC CGC CGG and CTG GGA TCC CTA GGA GGC GGA CCC TAG for mNaa12.

## Peptide competition assay

To determine the specificity of the mNaa12 antibody, a peptide competition assay was performed using the same peptide as used for immunization (aa 191–205: QENLAGGDSGSDGKD-C). 100 μg antibody were bound to 50 mg peptide-coupled CNBr-Sepharose (10 mg peptide/g Sepharose) in PBS + 0.2% Triton X-100 for 1 hr at 4°C on an orbital shaker. The beads were pelleted by centrifugation at 2.700 × g for 3 min at 4°C and 250 μL of the antibody-depleted supernatant diluted in 5 mL

TST for detection (1:100 final antibody dilution). Western blots of mouse lysates were probed with the depleted antibody or untreated antibody as control (1:100 dilution in TST).

## Cell lines
HEK293 cells were purchased from ATCC, authenticated via STR profiling, and confirmed mycoplasma free.

## Co-immunoprecipitation assay
Protein-protein interaction studies were performed in HEK293 cells. Briefly, $8 \times 10^5$ cells were seeded per well in 6-well plates. After 24 hr, cells were co-transfected with pcDNA3.1/V5-His-mNaa15 and p3xFLAG-CMV10-Naa10$^{235}$ (isoform 1), -Naa10$^{225}$ (isoform 2), -Naa11, or -Naa12 or the corresponding empty vectors. Cells were lysed after 48 hr in 200 µL PBS-X per well and cellular debris pelleted at 20.800 × g for 10 min at 4°C. 350 µL of the generated lysate was incubated with 1 µg anti-V5 antibody for 1 hr at 4°C, followed by a 30 min incubation with 30 µL protein-A Sepharose (Sigma-Aldrich). Protein complexes were washed three times by centrifugation (2.700 × g, 2 min) and eluted in 30 µL 2×SDS sample buffer.

Proteins were separated by SDS-PAGE and transferred onto a nitrocellulose membrane (Amersham Protran 0.2 µM NC) by immunoblotting. The membrane was blocked in 5% non-fat dry milk and incubated overnight with rabbit polyclonal anti-NAA15 (*Arnesen et al., 2005*) (1:2000, BioGenes) and rabbit monoclonal anti-NAA10 (anti-ARD1A, 1:1000, Cell Signaling, #13357) diluted in 1× PBS containing 1% non-fat dry milk and 0.1% Tween. The immunoblots were washed and incubated for 1 hr at room temperature (RT) with HRP-linked secondary antibody donkey anti-rabbit IgG (GE Healthcare, NA934). The HRP-signal was detected using SuperSignalTM West Pico PLUS Chemiluminescent Substrate Kit (Thermo Scientific) and ChemiDocTM XRS+ system (Bio-Rad) and visualized by ImageLab Software (Bio-Rad).

## Immunoprecipitation of Naa15 to form NatA complex
For immunoprecipitation of Naa15, 90–120 mg liver tissue from a WT- and Naa10 KO mouse was lysed in 500 µL IPH lysis buffer (50 mM Tris-HCl pH 8.0, 150 mM) NaCl, 5 mM EDTA, 0.5% NP-40, 1× complete EDTA-free protease inhibitor cocktail (Roche) using Kontes Pellet Pestle Motor and incubated on ice for 40 min. Cell debris was pelleted by centrifugation (17,000 × g, 4°C, 10 min) and the supernatants transferred to new Eppendorf tubes. The protein concentration was determined by BCA Protein Assay Kit (Thermo Scientific) and the tissue lysates were subsequently diluted with IPH lysis buffer to an equal protein concentration of 25 µg/µL. The WT- and Naa10 KO tissue lysates were then divided in two, whereof one half was mixed with 15 µg of anti-Naa15 antibody and the other half with 15 µg of anti-V5 antibody as a negative control. The mixtures were incubated at 4°C for 3 hr on a rotator. Afterwards, 180 µL of Protein A/G magnetic beads (Millipore) pre-washed in IPH lysis buffer was added to each sample and incubated overnight. Then, the magnetic beads were washed three times in IPH lysis buffer and two times in 1× acetylation buffer (100 mM Tris-HCl pH 8.5, 2 mM EDTA, 20% glycerol) prior to being resuspended in 90 µL of 2× acetylation buffer and used in a [$^{14}$C]-Ac-CoA-based acetylation assay.

## [$^{14}$C]-Ac-CoA-based acetylation assay of immunoprecipitated samples
Three positive replicates were prepared for each IP sample containing 10 µL IP beads, 200 µM synthetic oligopeptide SESS$_{24}$ (BioGenes), 100 µM [$^{14}$C]-Ac-CoA (Perkin-Elmer), and dH$_2$O to a final volume of 25 µL. In addition, two replicates for each IP sample were prepared without synthetic oligopeptide as negative controls. The samples were incubated at 37°C for 45 min in a thermomixer with shaking at 1400 rpm. Finally, the magnetic beads were isolated and 23 µL of the supernatant transferred to P81 phosphocellulose filter disks (Millipore). The filter disks were washed three times for 5 min in 10 mM HEPES buffer (pH 7.4) and air dried. To determine the amount of incorporated [$^{14}$C]-Ac, the filter disks were added to 5 mL Ultima Gold F scintillation mixture (Perkin-Elmer) and analyzed by a Perkin-Elmer TriCarb 2900TR Liquid Scintillation Analyzer.

## Proteomics sample preparation

Immunoprecipitation of Naa15 from a WT- and Naa10 KO mouse was performed as described above. Bound proteins were eluted from the magnetic beads using 60 µL of elution buffer (2% SDS, 100 mM Tris-HCl pH 7.6, 0.1 M DTT) and heated for 5 min at 95℃. The eluates were processed for LC-MS/MS analysis using filter-aided sample preparation (FASP) method (*Wiśniewski et al., 2009*). The eluted protein mixtures were mixed with UA buffer (8 M urea, 100 Mm Tris-HCl pH 8.0) and centrifuged through Microcon 30 kDa MWCO filters followed by Cys-alkylation with 50 mM iodoacetamide dissolved in UA buffer. Afterwards, the buffer was exchanged with 50 mM ammonium bicarbonate through sequential centrifugation, proteins were trypsinized (Sequencing Grade Modified Trypsin, Promega), and digested peptides were collected by centrifugation. Peptides were acidified using 5% formic acid and desalted using C18-stagetips according to protocol (*Rappsilber et al., 2007*). Briefly, 40 µg peptides from each sample were loaded onto C18-stagetips pre-conditioned with buffer A (1% formic acid). The C18-stagetips were then washed with buffer A, before peptides were eluted with buffer B (80% acetonitrile [ACN], 1% formic acid). The final eluate was concentrated by Speedvac to evaporate ACN and diluted to desired volume with 5% formic acid.

## Mass spectrometric analysis for immunoprecipitate

1 µg of the peptide samples were injected into an Ultimate 3000 RSLC system (Thermo Scientific) connected to a Q-Exactive HF mass spectrometer (Thermo Scientific) equipped with EASY-spray nano-electrospray ion source (Thermo Scientific). Trapping and desalting was performed with 0.1% TFA (flow rate 5 µL/min, 5 min) on a pre-column (Acclaim PepMap 100, 2 cm × 75 µm ID nanoViper column, 3 µm C18 beads). Peptides were separated on an analytical column (PepMap RSLC, 50 cm × 75 µm i.d. EASY-spray column, 2 µm C18 beads) during a biphasic ACN gradient with a flow rate of 200 nL/min. Solvent A (0.1% FA [vol/vol] in water) and B (100% ACN) were used for the following gradient composition: 5% B for 5 min, 5–8% B for 0.5 min, 8–24% B for 109.5 min, 24–35% B for 25 min and 35–80% B for 15 min, 80% B for 15 min, and conditioning with 5% B for 20 min. The mass spectrometer was operated in data-dependent mode to automatically switch between full-scan MS and MS/MS acquisition. MS spectra (m/z 375–1500) were acquired with a resolution of 120,000 at m/z 200, automatic gain control (AGC) target of $3 \times 10^6$, and maximum injection time (IT) of 100 ms. The 12 most intense peptides above an intensity threshold (50,000 counts, charge states 2–5) were sequentially isolated to an AGC target of $1 \times 10^5$ and maximum IT of 100 ms and isolation width maintained at 1.6 m/z, before fragmentation at a normalized collision energy of 28%. Fragments were detected in the orbitrap at a resolution of 15,000 at m/z 200, with first mass fixed at m/z 100. Dynamic exclusion was utilized with an exclusion time of 25 s and 'exclude isotopes' enabled. Lock-mass internal calibration (m/z 445.12003) was used. Raw files were processed with MaxQuant v. 1.6.17.0 (*Cox and Mann, 2008*) and searched against a database of Swiss-Prot annotated mouse protein sequences (retrieved 22.06.2018) in which the NAA12 sequence was added manually, and with a reverse decoy database. MaxQuant was run with default settings. Peptide and protein identifications were filtered to a 1% false discovery rate (FDR). Minimum peptide length was set to 7. Modifications included in protein quantification were oxidation (M), Nt-acetylation, acetylation (K), and phosphorylation (STY). Other parameters: match between runs – true, matching time window – 0.7 min, alignment time window – 20 min, find dependent peptides – true, mass bin size – 0.0065. Protein and peptide intensities were quantified by label-free quantification (LFQ) (*Cox et al., 2014*). The mass spectrometry proteomics data have been deposited to the ProteomeXchange Consortium via the PRIDE partner repository with the dataset identifier PXD026684.

## Whole-body CT scanning

CT scans were acquired on a Nanoscan PET/CT scanner from Mediso using Nucline v2.01 software. All mice were kept sedated under isoflurane anesthesia for the duration of the scan. Scans were acquired with an X-ray tube energy and current of 70 kVp and 280 uA, respectively. 720 projections were acquired per rotation, for three rotations, with a scan time of approximately 11 min, followed by reconstruction with a RamLak filter and voxel size 40 × 40 × 122 µm. For ex vivo analyses, mouse heads were fixed in 10% formalin buffered saline, followed by scanning and reconstruction with 1440 projections per revolution. Cranial volume was measured using VivoQuant software

(v2.50patch2) using the spline tool to manually and accurately draw around the circumference of the cranium on multiple stepwise 2D slices.

## Integrated N-terminal peptide enrichment (iNrich) assay

iNrich assays were performed as described (Ju et al., 2020). MEFs were made from E13.5 embryos, using standard techniques, with DMEM media supplemented with 10% fetal bovine serum (FBS), L-glutamine, and penicillin/streptomycin. Cells were harvested by trypsinization, washed twice with ice-cold phosphate-buffered saline (PBS, pH 7.4; Gibco), and resuspended in ice-cold lysis buffer containing 0.2 M EPPS (pH 8.0), 6 M guanidine, 10 mM TCEP (Thermo Fisher Scientific), and 40 mM 2-chloroacetamide (Sigma-Aldrich). After 10 min of incubation on 95℃, cells were lysed by ultrasonication by a BranSonic 400B. The proteins from the cell lysate were isolated by transferring supernatant after centrifugation at 12,000 g for 10 min at 4℃. The protein concentration of the collected supernatant was determined by bicinchoninic acid (BCA) protein assay. After enrichment of the N-terminal peptides, the peptide samples were analyzed by LC–MS/MS on an LTQ-Orbitrap XL mass spectrometer (Thermo Fisher Scientific) without further fractionation. Mass spectrometry data were uploaded to PRIDE under project name: Naa10 mutant mouse N-terminome LC-MS, project accession: PXD026410. Data analysis used unpaired, equal variance algorithm for Student's t-test.

## RNA and protein isolation and assays

70–120 mg tissues were lysed in 5 μL/mg tissue RIPA buffer (Sigma) with $1\times$ Complete protease inhibitors and 1 U/μL Superase In RNase inhibitor (Thermo Scientific) using Fisherbrand Pellet Pestle Cordless Motor. Afterwards, homogenization debris was removed by centrifugation at $20.800 \times$ g for 10 min at 4℃. Protein concentration was determined using APA assay (Cytoskeleton Inc) and 50 μg total protein were separated on SDS-PAGE followed by western blot. Membranes were stained with anti-Naa10, anti-Naa15, and anti-GAPDH antibodies (all Abcam).

For RNA purification, 30 μL clarified lysates were mixed with 70 μL RNase free water and RNA isolated using the RNeasy Mini Kit (Qiagen) according to the manufacturers recommendations, including on-column Dnase digest. 1 μg RNA was reverse transcribed using the TaqMan Reverse transcription kit and gene level detection performed using SYBR Green Master Mix (all Thermo Scientific). Relative expression was normalized to GAPDH and ACTB.

For the characterization of the mNaa12 antibody, tissue was lysed in 2 μL per mg tissue PBS-X (PBS + 0.2% [v/v] Triton X-100 + $1\times$ Complete protease inhibitor cocktail). 10–200 μg lysate were subjected to SDS-PAGE and western blot.

## Primers for mice qPCR

The following primers pairs were used: mNaa10-Exon2/3 F: 5′-ctcttggccccagctttctt-3′ and mNaa10-Exon3/4 R: 5′- tcgtctgggtcctcttccat-3′, mNaa11-F: 5′-accccacaagcaaagacagtg-3′ and mNaa11-R: 5′-agcgatgctcaggaaatgctct-3′, mNaa12(Gm16286)-F: 5′-acgcgtatgctatgaagcga-3′ and mNaa12 (Gm16286)-R: 5′-ccaggaagtgtgctaccctg-3′, mNaa15-F: 5′-gcagagcatggagaaaccct-3′ and mNaa15-R: 5′-tctcaaacctctgcgaacca-3′, mNaa50-F: 5′-taggatgccttgcaccttacc-3′ and mNaa50-R: 5′-gtcaatcgctgactcattgct-3′, mGAPDH-F: 5′-aggtcggtgtgaacggatttg-3′ and mGAPDH-R: 5′-tgtagaccatgtagttgaggtca-3′, mACTB-F: 5′-ggctgtattccctccatcg-3′ and mACTB-R: 5′-ccagttggtaacaatgccatgt-3′.

## Expression and purification of WT mouse, Naa10, and Naa12

All constructs were expressed in Rosetta (DE3)pLysS competent Escherichia coli cells. Cells were grown in LB-media to $OD_{600}$ 0.6–0.7 prior to inducing protein expression with 0.5 mM isopropyl β-D-1-thiogalactopyranoside (IPTG) at 18℃ for ~16 hr. All subsequent purification steps were carried out at 4℃. Cells were isolated by centrifugation and lysed in lysis buffer containing 25 mM Tris, pH 8.0, 150 mM NaCl, 10 mM β-mercaptoethanol (β-ME), 10 μg/mL phenylmethanesulfonylfluoride (PMSF), and DNase. The lysate was clarified by centrifugation and incubated with amylose agarose resin (New England Biolabs) for 1 hr before washing the resin with ≥100 column volumes of lysis buffer and then eluted with 10-column volumes of lysis buffer supplemented with 20 mM maltose. The resulting eluent was pooled and concentrated to ~10 mg/mL (30 kDa concentrator; Amicon Ultra, Millipore) such that 500 μL was loaded onto a Superdex 200 Increase 10/300 GL gel filtration column (GE Healthcare). The gel filtration run was performed in sizing buffer containing 25 mM

HEPES, pH 7.0, 200 mM NaCl, and 1 mM TCEP. After confirming the purity of the peak fractions at ~14 mL by denaturing SDS-PAGE (15% acrylamide), peak fractions were concentrated to 0.6 (6.1 µM) WT mouse Naa10 and 0.3 mg/mL (3.5 µM) WT mouse Naa12, as measured by UV$_{280}$ (Nanodrop 2000; Thermo Fisher Scientific), and stored at 4°C.

## Expression and purification of recombinant mNaa12 (1–160)-hNaa15 constructs

### Subcloning

Both full-length and truncated (1–160) mouse Naa12 were amplified from the pMAL-c5x Naa12 plasmid using Q5 HF Master Mix (NEB), AAAACCCGGGTATGAACATCCGCCGGGCTCGGC as the forward primer, and either AAAAGGTACCCTAGGAGGCGGACCCTAGGGTCTG (full-length) or AAAAGGTACCTCACCGTCTCAGCTCATCGGCCATCTG (1-160) as the reverse primer. An *Spodoptera frugiperda* (*Sf*9) pFastBac dual vector containing the sequence for the N-terminally 6xHis-tagged human Naa15 and truncated human Naa10 (residues 1–160) sequences was digested using KpnI-HF (NEB) and XmaI (NEB) to remove the human Naa10 sequence. The PCR product was also digested using the same restriction enzymes and ligated into the corresponding restriction sites using Mighty mix (Takara) using standard techniques. Both constructs were sequenced to validate the insert sequence and directionality.

Sf9 cells were grown to a density of 1 × 10$^6$ cells/ml and infected using the amplified baculoviruses to a multiplicity of infection (MOI) of ~1–2. Because the full-length mNaa12 construct did not produce protein, cells transfected with mNaa12$_{1-160}$/hNaa15 were grown at 27°C and harvested 48 hr post infection. All subsequent purification steps were carried out at 4°C. Following centrifugation of the cells, the pellet was resuspended and lysed in buffer containing 25 mM Tris, pH 8.0, 500 mM NaCl, 10 mM Imidazole, 10 mM β-ME, 10 µg/mL PMSF, DNase, and complete, EDTA-free protease inhibitor tablet (Roche). The lysate was clarified by centrifugation and incubated with nickel resin (Thermo Scientific) for 1 hr before washing the resin with ~125 column volumes of lysis buffer and then eluted with 10-column volumes of elution buffer (25 mM Tris, pH 8.0, 500 mM NaCl, 200 mM imidazole, 10 mM β-ME). Eluted protein was diluted to a final salt concentration of 200 mM NaCl and loaded onto a 5 mL HiTrap SP ion-exchange column (GE Healthcare). The protein was eluted in the same buffer with a salt gradient (200 mM to 1 M NaCl) over the course of 20 column volumes. Using the resulting peak fractions, the remainder of the purification was performed as described for the recombinant monomeric mNaa10 and mNaa12. However, resulting size-exclusion fractions were analyzed by denaturing SDS-PAGE using a 12% acrylamide gel, which was then silver stained (Bio-Rad) according to the manufacturer's instructions.

## In vitro radioactive acetyltransferase assays with recombinant protein

For recombinant mNaa12 and mNaa10 constructs, the assays were carried out in 40 mM HEPES, pH 7.5, 200 mM NaCl, where reactions were incubated with 150 nM of the gel-filtration purified WT mouse Naa10 or Naa12 in a 30 µL reaction volume containing each 250 µM substrate peptide and radiolabeled [$^{14}$C]acetyl-CoA (4 mCi/mmol; PerkinElmer Life Sciences) for 12 min (Naa12) or 13 min (Naa10) at 25°C. Respective time points were selected to ensure detection of sufficient activity within the linear range as determined by a time-course experiment. The substrate peptides used in the assay correspond to the first seven amino acids of β-actin (DDDIAAL-), γ-actin (EEEIAAL-), or the in vivo NatA complex substrate high-mobility group protein A1 (SESSS-), along with C-terminal positively charged residues for capture to the anion exchange paper. Background control reactions were performed in the absence of enzyme or in the absence of substrate peptide to ensure that any possible signal due to chemical acetylation was negligible. Each reaction was performed in triplicate.

To quench the reaction, 20 µL of the reaction mixture was added to negatively charged P81 phosphocellulose squares (EMD Millipore), and the paper disks were immediately placed in wash buffer (10 mM HEPES, pH 7.5). The paper disks were washed three times, at 5 min per wash, to remove unreacted acetyl-CoA. The papers were then dried with acetone and added to 4 mL of scintillation fluid, and the signal was measured with a PerkinElmer Life Sciences Tri-Carb 2810 TR liquid scintillation analyzer. The counts per minute were converted to molar units using a standard curve of known [$^{14}$C]acetyl-CoA concentrations in scintillation fluid.

Full peptide sequences:

β-actin: NH2-DDDIAALRWGRPVGRRRRPVRVYP-COOH
γ-actin: NH2-EEEIAALRWGRPVGRRRRPVRVYP-COOH
High-mobility group protein A1: NH2-SESSSKSRWGRPVGRRRRPVRVYP-COOH

For mNaa12-hNaa15, reactions were carried out similar to the monomeric mNaa12 and mNaa10, with the following exceptions: reactions were prepared by combining 21 μL of the respective fraction or sizing buffer with 5 μL of 10X buffer (500 mM HEPES, pH 7.5) to yield a buffer composed of 50 mM HEPES, pH 7.5, 140 mM NaCl, 0.7 mM TCEP, and 250 μM of each substrate upon reaction initiation. The reactions were allowed to incubate overnight at ambient temperatures (~25°C) and then quenched as described above. Control reactions were conducted in parallel as described above without conversion to molar units. Two technical replicates of the reactions were performed.

## Statistical analyses

Significant differences ($p < 0.05$) are indicated by asterisks. Weight analyses were performed using generalized estimating equations (GEEs) (*Zeger and Liang, 1986*), an extension of generalized linear models that adjusts for the effects of autocorrelation resulting from multiple measurements, and implemented within version 15.1 of Stata (StataCorp 2017).

## Genotype distribution analyses and modeling

Genotype distributions for several *Naa10/Naa12* KO crosses were analyzed and models were created to estimate the number of the live (or at least intact) embryos or pups that are expected to be observed based on the assumptions and rules that follow. (1) Genotype survival rates are the fractional value, from 0 to 1 (or 0–100%), of the expected Mendelian fraction for that genotype in the cross being evaluated. (2) Genotype survival rates cannot exceed 1 (or 100%). (3) Genotype survival rates can decrease with age but not increase. (4) WT genotypes (*Naa10$^{+/Y}$*; *Naa12$^{+/+}$* and *Naa10$^{+/+}$*; *Naa12$^{+/+}$*) are expected to have 100% survival at all ages because the models predict the number of embryos or pups relative to WT survival. Reductions in overall in litter sizes for crosses were estimated through other calculations. (5) The biological basis for a reduced survival rate assumes that loss of one or more copies of either *Naa10* or *Naa12* removes or reduces functions that are required for successful embryonic development or postnatal life. Reduced survival rates for non-WT genotypes were estimated based on differences (delta) between the expected number of embryos or pups based on the Mendelian proportion (or the current best model) and the observed number of embryos or pups. Separate comparisons were made using deltas for each specific age and for the cumulative numbers at each age. (6) Genotype frequencies for each model were calculated as described in the section below. (7) The fit between a model and the observed data was determined by calculating the relative standard deviation (SD) for the deltas across all genotypes, for example, the SD across genotypes divided by the number of animals observed (either age-specific or cumulative). Each model was evaluated at each age by minimizing the relative SD for all genotypes at that age and over all ages. The final model ($D_4$) was created by refining the assumptions for model $D_3$ in a sequential series of comparisons of survival rates for genotypes 12, 11, 6, 10, and 5 in that order.

## Genotype frequency calculations

The models described adjust the expected observed genotype frequencies at each age to account for loss of embryos or pups due to the presumed lethal effects of one or more genotypes. The models account directly for the effect of genotype-specific mortality by reducing the number (or frequency) observed for that genotype in the sample and thus increasing the expected proportion of other genotypes. This also indirectly implies a larger theoretical litter size at conception, which can be used to determine the theoretical litter sizes had there been no mortality in the affected genotypes. The predicted proportion for each genotype is calculated at each age as the genotype Mendelian frequency multiplied by the fractional genotype survival at that age divided by the expected total fractional survival (i.e., one minus the sum of all genotype fractional losses). The formula is

For all 'n' possible genotypes:

$$G_x = M_x * S_x / (1 - -[(1 - -S_1) * M_1 + (1 - -S_2) * M_2 + \ldots + (1 - -S_n) * M_n])$$

where

$G_x$ = model genotype fractional value (frequency) for genotype 'x' ($G_x$ value from 0 to 1),
$M_x$ = Mendelian fractional value (frequency) for genotype 'x' for the cross, and
$S_x$ = fractional survival between 0 and 1.

$(1 - S_n) * M_n$ is the fractional reduction due to survival < 100% for genotype 'n', for example, when $S_n = 1$ (e.g., 100% survival), the loss is zero; when $S_n = 0$ (e.g., 0% survival), the loss is $M_n$ or the entire Mendelian fraction.

Note that the sum of all $G_x$ for all genotypes at any age is always equal to 1 (or 100%).

## Acknowledgements

GTO thanks Huiju Jo for supporting mouse genotyping. GJL would like to thank Ryan Driscoll for mouse genotyping, Fatima Inusa for embryo dissection, Jemima Kadima for mouse husbandry, Leyi Li for transgenic mouse assistance, Melissa Nashat, Jodi Coblentz, Rachel Rubino, and Lisa Bianco for veterinary advice, and Vicky Brandt for editorial assistance.

This work was supported by National Research Foundation of Korea (NRF) grants funded by the Korean Government (2020R1A3B2079811 (GO), 2017RIDIAB03032286 (ML), and 2020RICIC1007686 (ML)). Research reported in this publication was also supported by the National Institute of General Medical Sciences of the National Institutes of Health (NIH) under Award Numbers R35GM133408 (GJL) and R35GM118090 (RM), and NIH grant HL148165 (SJC). The work was also supported by the Research Council of Norway (Project 249843), the Norwegian Health Authorities of Western Norway (projects 912176 and F-12540-D11382), and the Norwegian Cancer Society (PR-2009-0222). The content is solely the responsibility of the authors and does not necessarily represent the official views of the National Institutes of Health. Funding was also provided by the Stanley Institute for Cognitive Genomics at Cold Spring Harbor Laboratory, the George A Jervis Clinic and the Department of Human Genetics, Laboratory of Genomic Medicine at the New York State Institute for Basic Research in Developmental Disabilities (IBR), New York State Office for People with Developmental Disabilities. Part of the work was carried out at the Proteomics Unit at University of Bergen (PROBE).

## Additional information

### Funding

| Funder | Grant reference number | Author |
| --- | --- | --- |
| National Research Foundation of Korea | 2020R1A3B2079811 | Goo Taeg Oh |
| National Research Foundation of Korea | 2017RIDIAB03032286 | Mi-Ni Lee |
| National Research Foundation of Korea | 2020RICIC1007686 | Mi-Ni Lee |
| National Institute of General Medical Sciences | R35GM133408 | Gholson J Lyon |
| National Institute of General Medical Sciences | R35GM118090 | Ronen Marmorstein |
| Research Council of Norway | Project 249843 | Thomas Arnesen |
| National Institutes of Health | Project 249843 | Simon J Conway |
| Norwegian Health Authorities of Western Norway | 912176 | Thomas Arnesen |
| Norwegian Health Authorities of Western Norway | F-12540-D11382 | Thomas Arnesen |
| Norwegian Cancer Society | PR-2009-0222 | Thomas Arnesen |

The funders had no role in study design, data collection and interpretation, or the decision to submit the work for publication.

## Author contributions

Hyae Yon Kweon, Formal analysis, Investigation, Writing - original draft, Writing - review and editing; Mi-Ni Lee, Supervision, Investigation, Project administration, Writing - review and editing; Max Dorfel, Seungwoon Seo, Thomas PaPazyan, Nina McTiernan, Rasmus Ree, Andrew Garcia, Michael Flory, Jonathan Crain, Alison Sebold, Scott Lyons, Ahmed Ismail, Elaine Marchi, Seong-keun Sonn, Se-Jin Jeong, Sejin Jeon, Shinyeong Ju, Simon J Conway, Taesoo Kim, Hyun-Seok Kim, Cheolju Lee, Tae-Young Roh, Investigation; Leah Gottlieb, Investigation, Writing - review and editing; David Bolton, Formal analysis, Writing - review and editing; Thomas Arnesen, Ronen Marmorstein, Supervision, Writing - review and editing; Goo Taeg Oh, Funding acquisition, Project administration, Writing - review and editing; Gholson J Lyon, Conceptualization, Formal analysis, Supervision, Funding acquisition, Investigation, Writing - original draft, Project administration, Writing - review and editing

## Author ORCIDs

Hyae Yon Kweon (iD) https://orcid.org/0000-0002-1534-0767
Nina McTiernan (iD) http://orcid.org/0000-0002-1749-6933
Se-Jin Jeong (iD) http://orcid.org/0000-0002-6375-5334
Sejin Jeon (iD) http://orcid.org/0000-0002-3819-3421
Shinyeong Ju (iD) http://orcid.org/0000-0001-5483-4690
Taesoo Kim (iD) http://orcid.org/0000-0002-3902-1058
Cheolju Lee (iD) http://orcid.org/0000-0001-8482-4696
Tae-Young Roh (iD) http://orcid.org/0000-0001-5833-0844
Ronen Marmorstein (iD) http://orcid.org/0000-0003-4373-4752
Goo Taeg Oh (iD) https://orcid.org/0000-0002-1104-1698
Gholson J Lyon (iD) https://orcid.org/0000-0002-5869-0716

## Ethics

Animal experimentation: All experiments were performed in accordance with guidelines of International Animal Care and Use Committee (IACUC) of Ewha Womans University (protocol #18-012), Cold Spring Harbor L aboratory (CSHL) protocol # 579961 18, and Institute for Basic Research in Developmental Disabilities (IBR) (protocol #456).

## Decision letter and Author response

Decision letter https://doi.org/10.7554/eLife.65952.sa1
Author response https://doi.org/10.7554/eLife.65952.sa2

## Additional files

### Supplementary files

• Supplementary file 1. Tables related to genotypes, phenotypes and Mendelian ratios. (**a**) Genotypes of offspring from $Naa10^{+/-}$ female mice crossed to the $Naa10^{+/Y}$ male mice. Expected and observed Mendelian ratio of genotypes in offspring at E10.5, E13.5, E18.5 and adults from crosses of $Naa10^{+/-}$ female and $Naa10^{+/Y}$ male mice. The percentage of adult $Naa10^{-/Y}$ mice significantly decreases. (**b**) Genotypes of offspring from $Naa10^{+/tm1a}$ female mice crossed to the $Naa10^{+/Y}$ male mice. Expected and observed Mendelian ratio of genotypes in offspring at E10.5, E13.5, E18.5 and adults from crosses of $Naa10^{+/tm1a}$ female and $Naa10^{+/Y}$ male mice. The percentage of adults $Naa10^{tm1a/Y}$ mice significantly decreases. (**c**) Cervical fusion skeletal analyses in $Naa10$ knockout (KO) mice. (**d**) Matings and litter size analyses. (**e**) Genotypes of offspring from $Naa12^{+/-}$ female mice crossed to the $Naa12^{+/-}$ male mice. Expected and observed Mendelian ratio of genotypes in offspring from crosses. (**f**) Genotypes of offspring from $Naa10^{+/-}$ $Naa12^{+/+}$ female mice crossed to the $Naa10^{+/y}$ $Naa12^{+/-}$ male mice. Expected and observed Mendelian ratio of genotypes in offspring from crosses. (**g**) Genotypes of offspring from $Naa10^{+/-}$ $Naa12^{+/+}$ female mice crossed to the $Naa10^{+/y}$ $Naa12^{-/-}$ male mice. Expected and observed Mendelian ratio of genotypes in offspring from crosses. (**h**) Mendelian and observed offspring distributions from Naa10(+/Y); Naa12(+/-) male and Naa10(+/-); Naa12(+/-) female breeding. (**i**) Mendelian and observed offspring distributions

from Naa10(+/Y); Naa12(-/-) male and Naa10(+/-); Naa12(+/-) female breeding. (**j**) Mendelian and observed postnatal offspring distributions from Naa10(+/Y); Naa12(+/-) male and Naa10(+/-); Naa12 (+/+) female breeding. (**k**) Mendelian and observed postnatal offspring distributions from Naa10(+/Y); Naa12(-/-) male and Naa10(+/-); Naa12(+/+) female breeding. (**l**) Mendelian and observed age-specific offspring distributions from four crosses. (**m**) Mendelian and observed cumulative offspring distributions from all four crosses. (**n**) Mice analyzed by weighing, according to genotype. (**o**) Effects of *Naa10* KO on growth rate of *Naa10* mice on pure genetic background. (**p**) Effects of *Naa10* and *Naa12* KOs on growth rate on mixed genetic background.

• Supplementary file 2. Mass spectrometry analyses. (**a**) N-termini detected in mouse embryonic fibroblasts (MEFs) (Excel sheet). (**b**) N-termini detected in all MEF samples (Excel sheet). (**c**) Header Key, showing the abbreviation meanings for (**a**) and (**b**) (Excel sheet). (**d**) Mass spectrometry of immunoprecipitated Naa15 complexes (Excel sheet).

• Transparent reporting form

## Data availability

All data generated or analysed during this study are included in the manuscript and supporting files. Mass spectrometry data were uploaded to PRIDE under Project Name: Naa10 mutant mouse N-terminome LC-MS, Project accession: PXD026410 and Project Name: mNaa10-KO liver immunoprecipitation, Project accession: PXD026684.

The following datasets were generated:

| Author(s) | Year | Dataset title | Dataset URL | Database and Identifier |
|---|---|---|---|---|
| Ju S, Lee C | 2021 | Naa10 mutant mouse N-terminome LC-MS | https://www.ebi.ac.uk/pride/archive/projects/PXD026410 | PRIDE, PXD026410 |
| Ree R, Arnesen T | 2021 | mNaa10-KO liver immunoprecipitation | https://www.ebi.ac.uk/pride/archive/projects/PXD026684 | PRIDE, PXD026684 |

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
