## [Decision Letter]

**Acceptance summary:**

The manuscript describe the phenotype when Naa10 and Naa12 are inactivated and how Naa12 compensates Naa10 deficiency, reducing the phenotype observed when Naa10 is inactivated. The finding of a new N-acetyltransferase will be useful to the field of N-terminal acetylation.

**Decision letter after peer review:**

Thank you for submitting your article "Naa12 rescues embryonic lethality in Naa10-Deficient Mice in the amino-terminal acetylation pathway" for consideration by *eLife*. Your article has been reviewed by 3 peer reviewers, and the evaluation has been overseen by a Reviewing Editor and Philip Cole as the Senior Editor. The reviewers have opted to remain anonymous.

Summary:

Mutations in the X-linked gene Naa10, which encodes a major protein N-terminal acetyltransferase, are known to be causative in Ogden syndrome, a genetic disorder associated with infantile death. The manuscript reports that hemizygous knockout of Naa10 in male mice does not cause complete embryonic lethality despite the fact that it is responsible for the acetylation of about 50% of all the proteins. This leads the authors to discover a paralogous mouse gene Naa12 and demonstrate that Naa12 can compensate for Naa10 loss of function and that null mutations in both genes lead to complete embryonic lethality. This is an important finding. The authors also provided a thorough account of the variety of development abnormalities in the Naa10 hemizygous mice at all stages of development. Importantly, the authors identified several phenotypes in the mice that upon further analysis, we also noted in the some patients.

All reviewers are enthusiastic about the finding of a new N-terminal acetyltransferase and think that this is a major strength of the manuscript. In addition, the phenotypical characterization of Naa10 is also useful to the field.

Essential revisions:

Some of the revision recommendations requires additional experiments. The reviewers and the Reviewing Editor have discussed and agree that these additional experiments should only take a few weeks to complete.

1) Carry out an in vitro acetyltransferase assays with Naa12-Naa15 to validate the proposed mechanistic model.

2) Check whether Pax3 and HoxC8 proteins' abundance/stability are affected in NAA10 KO mouse tissue or MEF.

3) Obtain experimental evidence to support that N-terminal acetylation is lost in the double knockouts. If such evidence is not possible or will take too much experimental effort, please tone down the claim that the complete machinery for the process of amino-terminal acetylation of proteins in mouse development is identified.

4) Revise the text according to the comments by all three reviewers to make it easier for readers to understand.

*Reviewer #1 (Recommendations for the authors):*

1) It is important to show that N-terminal acetylation is lost in the double knockouts. Only with that, the authors can conclude that they have identified the "the complete machinery for the process of amino-terminal acetylation of proteins in mouse development."

2) Naa12 is new, so if not done yet, the sequence needs to be deposited into Genbank.

3) The presentation needs to be polished.

i) The title "Naa12 rescues embryonic lethality in Naa10-Deficient 1 Mice in the amino-terminal acetylation pathway" is misleading. When I saw the title, I got the impression that Naa10-dficient 1 mice show embryonic lethality. I would suggest to change it to indicate that Naa10 and Naa12 have redundant roles in embryonic development. Also, "Naa10-Deficient 1 Mice" needs to be changed to "Naa10-deficient mice."

ii) In the impact statement "Mice doubly deficient for Naa10 and Naa12 display embryonic lethality…", the word "doubly " is unnecessary.

iii) Too many acronyms, which make the reading a bit difficult. The terms NTA and Nt-acetylation could be avoided.

iv) At the end of page 9, please cite the sequence alignment in Figure S6.

v) On page 12, "Naa12 may rescue loss of Naa10 in mice" could be more assertive.

vi) Overall, I feel that the authors could polish the manuscript so that the salient points could be conveyed more easily to readers.

*Reviewer #2 (Recommendations for the authors):*

Overall this is a very good paper, but some issues need further attention.

– Authors present data showing that mNAA12 can acetylate some peptides at their N-end, however these reactions are performed with just the monomer and NatA activity is catalysed in vivo by the heterodimer Naa10-Naa15, showing an N-terminal acetylation in vitro activity different to Naa10 and being more similar to the observed in cells. Therefore, it is important to characterize Naa12-Naa15 in vitro enzymatic activity assessing its specificity using diverse NatA described peptidic substrates and compare it with the regular NatA enzymatic complex that presents NAA10 as catalytic subunit.

– There is a clear phenotype when mNAA10 is inactivated however, there is not a dramatic defect in protein N-terminal acetylation as it has been observed in other organisms like yeast and Arabidopsis thailiana. It has been observed by several authors that some proteins stability is affected when they are not N-terminal acetylated influencing the biological activity of these proteins. Authors comment that Pax3 and HoxC8 proteins regulate some of the biological activities deregulated in mice when mNAA10 is inactivated. As both proteins are NatA substrates and their stability could be affected when mNAA10 is inactivated it would be interesting to assess their abundance in NAA10KO embryos, mice tissue and/or MEFs in comparison with the observed in wild-type samples. These results could bring some light to the molecular mechanisms that are governing the effects observed in these mice.

*Reviewer #3 (Recommendations for the authors):*

Line 55 of the abstract "Male mice lacking Naa10 show no globally apparent in vivo NTA impairment and, surprisingly, do not exhibit embryonic lethality". This statement is misleading as written (and appears again in the discussion) as later in the paper, the authors describe that homozygous mice were underrepresented at birth, and describe embryonic phenotypes and embryos that underwent lysis by 10.5. Suggest rephrasing to.…do not exhibit complete embryonic lethality.

The fate of the Naa10 mice could be written more clearly. As it stands, there is a vagueness as to the number of mice that were not born, that die as early neonates and the number that survived to adulthood. Authors should consider adding a Kaplan Myer survival curve. Statements like "Some of the Naa10 hemizygous mice survived to 4 months is.." is not helpful.

Given the phenotypic effect of Naa12+/- Naa10 +/-, females, it may have been interesting to access any effects in acetylation?

---

## [Author Response]

Essential revisions:Some of the revision recommendations requires additional experiments. The reviewers and the Reviewing Editor have discussed and agree that these additional experiments should only take a few weeks to complete.1) Carry out an in vitro acetyltransferase assays with Naa12-Naa15 to validate the proposed mechanistic model.

We added these analyses in the manuscript, and this does add new evidence showing that the complex has activity.

2) Check whether Pax3 and HoxC8 proteins' abundance/stability are affected in NAA10 KO mouse tissue or MEF.

We have attempted to perform the experiments, but the results were not compelling, as described below.

3) Obtain experimental evidence to support that N-terminal acetylation is lost in the double knockouts. If such evidence is not possible or will take too much experimental effort, please tone down the claim that the complete machinery for the process of amino-terminal acetylation of proteins in mouse development is identified.

Double knockouts show early embryonic lethality (Table 4), so if we want to confirm the N-terminal acetylation level of double knockout, the embryos have to be isolated before E8.5 day. We have not yet been able to obtain sufficient material for these experiments. Therefore, we toned down the wording regarding complete machinery, as recommended.

4) Revise the text according to the comments by all three reviewers to make it easier for readers to understand.

We have made these changes.

Reviewer #1 (Recommendations for the authors):1) It is important to show that N-terminal acetylation is lost in the double knockouts. Only with that, the authors can conclude that they have identified the "the complete machinery for the process of amino-terminal acetylation of proteins in mouse development."

We have decided to remove this sentence, as the reviewer is correct that there might be even more unidentified N-terminal acetyltransferase enzymes in mice. For now, we simply present the data showing embryonic lethality for mice doubly deficient in Naa10 and Naa12. We did add a concluding sentence to the abstract, which is more toned down.

2) Naa12 is new, so if not done yet, the sequence needs to be deposited into Genbank.

When looking for homolog sequences, we did a blast-search using the sequence for Naa10. That led to the automatically annotated gene identifier on chromosome 18, which we named Naa12: Gm16286. This predicted gene translates to NM_001384178.1 (predicted gene mRNA) and NP_001371107.1 (N-α-acetyltransferase 11-like protein). This means that Naa12 is already in Genbank, however, with a different name and the status "predicted". We cannot find a way for us to re-annotate existing sequences, so we assume it will be re-annotated by the curators of Genbank after our paper is published. We modified the following sentence to read: " We found a predicted gene (Gm16286, UniProt: Q9CQX6) on chromosome 18, with high similarity to Naa10, which we name Naa12, and RiboSeq and mRNA traces of this region suggest possible transcription and translation of this gene.”

3) The presentation needs to be polished.

We have edited throughout with this in mind.

i) The title "Naa12 rescues embryonic lethality in Naa10-Deficient 1 Mice in the amino-terminal acetylation pathway" is misleading. When I saw the title, I got the impression that Naa10-dficient 1 mice show embryonic lethality. I would suggest to change it to indicate that Naa10 and Naa12 have redundant roles in embryonic development. Also, "Naa10-Deficient 1 Mice" needs to be changed to "Naa10-deficient mice."

We have changed the title to: *Naa12* compensates for *Naa10* in mice in the amino-terminal acetylation pathway.

ii) In the impact statement "Mice doubly deficient for Naa10 and Naa12 display embryonic lethality…", the word "doubly " is unnecessary.

Removed.

iii) Too many acronyms, which make the reading a bit difficult. The terms NTA and Nt-acetylation could be avoided.

We replaced NTA and Nt-acetylation with amino-terminal acetylation throughout the manuscript. We have also tried throughout to expand on acronyms.

iv) At the end of page 9, please cite the sequence alignment in Figure S6.

We inserted this.

v) On page 12, "Naa12 may rescue loss of Naa10 in mice" could be more assertive.

We changed this to "*Naa12* rescues loss of *Naa10* in mice".

vi) Overall, I feel that the authors could polish the manuscript so that the salient points could be conveyed more easily to readers.

We have edited the manuscript throughout, attempting to clarify reading.

Reviewer #2 (Recommendations for the authors):Overall this is a very good paper, but some issues need further attention.– Authors present data showing that mNAA12 can acetylate some peptides at their N-end, however these reactions are performed with just the monomer and NatA activity is catalysed in vivo by the heterodimer Naa10-Naa15, showing an N-terminal acetylation in vitro activity different to Naa10 and being more similar to the observed in cells. Therefore, it is important to characterize Naa12-Naa15 in vitro enzymatic activity assessing its specificity using diverse NatA described peptidic substrates and compare it with the regular NatA enzymatic complex that presents NAA10 as catalytic subunit.

We have attempted to prepare recombinant Naa12/Naa15 complex from Sf9 insect cells for biochemical studies, but were unfortunately unable to prepare sufficient levels of complex for detailed enzymatic analysis. Nonetheless, we were able to prepare a sufficient amount of recombinant complex to analyze fractions from a final size exclusion chromatograph purification. This analysis demonstrated that the Naa12/Naa15 complex harbored detectable amino-terminal acetylation activity toward the SESSSKS- peptide (Figure 4C), thus demonstrating catalytic activity of a NatA complex with mouse Naa12.

– There is a clear phenotype when mNAA10 is inactivated however, there is not a dramatic defect in protein N-terminal acetylation as it has been observed in other organisms like yeast and Arabidopsis thailiana. It has been observed by several authors that some proteins stability is affected when they are not N-terminal acetylated influencing the biological activity of these proteins. Authors comment that Pax3 and HoxC8 proteins regulate some of the biological activities deregulated in mice when mNAA10 is inactivated. As both proteins are NatA substrates and their stability could be affected when mNAA10 is inactivated it would be interesting to assess their abundance in NAA10KO embryos, mice tissue and/or MEFs in comparison with the observed in wild-type samples. These results could bring some light to the molecular mechanisms that are governing the effects observed in these mice.

We included the references to Pax3 and Hoxc8 as possible future directions, and it would be an entire new paper to demonstrate this mechanism. Nonetheless, to attempt to address this for the reviewer, we have performed Western blotting for Pax3 and Hoxc8. It is not easy to detect Pax3 level with western blot, so B16F10 (mouse melanoma cell line) stably expressing pax3 was used as a control. We have detected a slight decline only in NAA10KO E10.5 embryos, but this is very small and likely not significant, and this is only semi-quantitative at best with Western blotting, and so we are not prepared to publish this result until we have confirmed it with further experiments, including mass spectrometry. For Hoxc8, signal detection was not successful using one antibody that we purchased, and we will have to try again for a future paper. Of course, this does not specifically address the N-terminal acetylation of either protein, but N-terminal peptides from Pax3 and Hoxc8 are not present in our mass spectrometry data (included here and also unpublished data). As such, we cannot determine if there are differences in their N-terminal acetylation levels, and we are beginning now various approaches to try to determine this.

Reviewer #3 (Recommendations for the authors):Line 55 of the abstract "Male mice lacking Naa10 show no globally apparent in vivo NTA impairment and, surprisingly, do not exhibit embryonic lethality". This statement is misleading as written (and appears again in the discussion) as later in the paper, the authors describe that homozygous mice were underrepresented at birth, and describe embryonic phenotypes and embryos that underwent lysis by 10.5. Suggest rephrasing to.…do not exhibit complete embryonic lethality.

We modified this to read: "Male mice lacking *Naa10* show no globally apparent in vivo amino-terminal acetylation impairment and do not exhibit complete embryonic lethality."

The fate of the Naa10 mice could be written more clearly. As it stands, there is a vagueness as to the number of mice that were not born, that die as early neonates and the number that survived to adulthood. Authors should consider adding a Kaplan Myer survival curve.

We have added such a curve, and have referred to it in the Results.

Statements like "Some of the Naa10 hemizygous mice survived to 4 months is.." is not helpful.

We have quantified this in the Results, with statements such as: “Out of 32 Naa10-/Y that survived past the third day of life and which were then examined longitudinally, about 60% survived past 200 days of life (~7 months)”.

Given the phenotypic effect of Naa12+/- Naa10 +/-, females, it may have been interesting to access any effects in acetylation?

This is not an easy experiment to perform, as we have spent several years attempting to demonstrate decreased acetylation in KO males, with no success, leading us to the characterization of Naa12. The compound heterozygous Naa12+/- Naa10 +/- females are born in small numbers, and we have now quantified this in detail in a new section titled “Genotype distribution modeling of Naa10- and Naa12-deficient offspring”. It is outside the scope and timeframe for the current paper to attempt to quantify amino-terminal acetylation levels in the compound heterozygous females, and we are likely going to focus instead on dying (or just before dying) doubly deficient embryos, where the degree of acetylation difference will likely be greater.